# Satellite-based Analysis of Ocean-Surface Stress across the Ice-free and Ice-covered Polar Oceans

Chao Liu[1], Lisan Yu[1]

[1]Woods Hole Oceanographic Institution, Woods Hole, MA, USA

*Correspondence to*: Chao Liu (chao.liu@whoi.edu)

**Abstract.** Ocean-surface stress is a critical driver of polar sea ice dynamics, air-sea interactions, and ocean circulation. This work provides a daily analysis of ocean-surface stress on 25-km Equal-Area Scalable Earth (EASE) Grids across the ice-free and ice-covered regions of the polar oceans (2011-2018 for Arctic, 2013-2018 for Antarctic), covering latitudes north of 60°N in the Arctic and south of 50°S in the Antarctic and Southern Ocean. Ocean-surface stress is calculated using a bulk parameterization approach that combines ocean-surface winds, ice motion vectors, and sea surface height (SSH) data from multiple satellite platforms. The analysis captures significant spatial and temporal variability in ocean-surface wind stress and the resultant wind-driven Ekman transport, while providing enhanced spatiotemporal resolution. Two sensitivity analyses are conducted to address key sources of uncertainty. The first addresses the fine-scale variability in SSH fields, which was mitigated using a 150-km Gaussian filter to smooth three-day SSH datasets and enhance compatibility with the other monthly product, followed by linear interpolation to achieve daily resolution. The second investigates uncertainty in the ice-water drag coefficient, which revealed that variations in the coefficient have a proportional influence on the computed ocean-surface stress under the tested conditions. These uncertainties are most pronounced during winter, with median values reaching 20% in the Arctic and 40% in the Southern Ocean. Validation efforts using Ice-Tethered Profiler velocity records revealed weak to moderate correlations with satellite-derived stress (r = 0.4–0.8) between observed surface velocities and satellite-derived estimates (Ekman + geostrophic) at daily resolution, with significantly improved agreement when averaged to weekly means. This dataset is publicly available at https://doi.org/10.5281/zenodo.14750492 (Liu & Yu, 2024).

## 1 Introduction

Earth's polar regions have undergone profound changes over the past decades, with sea ice playing a central role in the polar climate system. By modulating heat, momentum, and freshwater exchanges at the atmosphere-ice-ocean boundary, sea ice directly influences global climate dynamics (Meehl, 1984; Stammerjohn et al., 2012). In the Arctic, rapid sea ice decline has transitioned the region from predominantly thick, multiyear ice to thinner, more dynamic ice, with increased interannual variability (Comiso et al., 2008; Stroeve and Notz, 2018; Moore et al., 2022; Babb et al., 2022). Meanwhile, Antarctic sea ice trends have shown greater complexity, with a modest long-term increase observed until the mid-2010s, followed by a record loss in 2017 and a subsequent continued decline (Liu et al., 2004; Parkinson, 2019; Turner et al., 2022; Purich & Doddridge, 2023). These changes in sea ice extent and thickness have significant implications for polar systems and global climate feedbacks, influencing the Arctic's ability to regulate planetary heat, as well as impacting marine ecosystem, carbon cycling, nutrient distribution, and thermohaline circulation (Talley, 2013; Campbell et al., 2019).

Atmospheric circulation is a primary driver of sea ice dynamics and variability. Geostrophic winds, for instance, account for over 70% of sea ice velocity variability (Thorndike and Colony, 1982; Maeda et al., 2020), while broader climate modes, including the Arctic Oscillation, Pacific Decadal Oscillation, and Southern Annular Mode, influence ice extent and distribution (Rigor et al., 2002; Park et al., 2018; Lefebvre et al., 2004). These wind-driven processes interact with sea ice to modify ocean-surface stress, impacting Ekman dynamics and the transport of heat, salt, and nutrients (Yang, 2006, 2009; Meneghello et al., 2018). This feedback mechanism, often described as the "ice-ocean governor" (Meneghello et al., 2017), plays an important role in regulating polar freshwater storage and circulation (Marshall and Speer, 2012; Abernathey et al., 2016; Ma et al., 2017).

Surface stress plays a pivotal role in driving Arctic Ocean circulation by mediating the transfer of momentum from the atmosphere to the ocean. In the Arctic, sea ice acts as a modulator of this momentum exchange, either dampening or amplifying the transfer depending on its concentration and mechanical properties. Recent projections indicate that as the Arctic climate warms, sea ice will become thinner and less extensive, leading to a more efficient transfer of wind energy to the ocean surface (Muilwijk et al., 2024). This enhanced momentum transfer is expected to accelerate surface currents, increase ocean kinetic energy, and intensify vertical mixing processes (Martin et al. 2014; Martin et al. 2016). However, current climate models exhibit considerable uncertainty in simulating these processes due to simplified representations of atmosphere-ice-ocean interactions. Therefore, developing observationally based surface stress products is essential for validating and improving model simulations, leading to more accurate predictions of future Arctic Ocean dynamics and their global implications.

To address the complexity of ice–ocean interactions, recent modeling advances have highlighted the pivotal role of sea ice form drag in governing momentum exchange at the ocean–ice–atmosphere interface. Tsamados et al. (2014) introduced a physically grounded parameterization of ice form drag that accounts for ice morphological features—such as ridges, floe edges, and melt pond geometry—and demonstrated that spatial and temporal variability in drag coefficients can substantially influence sea ice dynamics and the spin-up of the Arctic Ocean. Extending this approach, Sterlin et al. (2023) implemented a variable ice form drag scheme in the NEMO-LIM3 ocean–sea ice model and found that it exerts a pronounced control over

ocean surface stress patterns, mixed layer depth, sea surface salinity, and upper ocean temperature across both polar regions.
These modeling efforts reveal that ice form drag is not merely a secondary detail, but a first-order process in polar ocean
circulation and surface forcing. However, the representation of ice–ocean drag—often quantified through the coefficient—
remains highly uncertain, as it can vary markedly with environmental conditions including ice concentration, surface roughness,
and the presence of waves (Lüpkes & Gryanik, 2015; Brenner et al., 2021). This highlights the growing need for
observationally based estimates of ocean surface stress that can support parameterization efforts, constrain model behavior,
and improve the physical realism of coupled ocean–ice simulations.
Despite significant advancements in understanding these processes, direct measurements at the ice-ocean interface remain
limited, with most data concentrated in the Arctic's Canada Basin (Smith et al., 2019; Regan et al., 2019). Satellite remote
sensing has been instrumental in addressing these gaps, providing open ocean-surface wind retrievals available since 1988 (Yu
& Jin, 2014a) and tracking sea ice motions since 1978 (Cavalieri et al., 1996). Recent advances in satellite altimetry further
enable high-resolution monitoring of sea surface height (SSH) changes, offering new insights into mesoscale ocean dynamics
(Armitage et al., 2016, 2017; Prandi et al., 2021).
Building upon the concepts developed in previous studies (Yang, 2006, 2009; Meneghello et al., 2018), this analysis utilizes
recent satellite-based datasets on wind, ice motion, and SSH to analyze ocean-surface stress across both ice-free and ice-
covered polar seas. Specifically, we present a daily analysis of ocean-surface stress at 25-km resolution using Equal-Area
Scalable Earth (EASE, see glossary in Table A1 for more details) Grids from 2011 to 2018 for Arctic and 2013-2018 for
Antarctic, covering latitudes north of 60°N in the Arctic and south of 50°S in the Antarctic and Southern Ocean (Figure 1).
Section 2 provides a description of the satellite datasets used and processing steps, along with the methods for calculating
ocean-surface stress and Ekman circulation. Section 3 presents the time-mean patterns and variability of the derived surface
stress and Ekman pumping fields. Section 4 addresses quantification of uncertainties in the analysis, including sensitivity to
the ice-water drag coefficient and comparisons of with in-situ data.


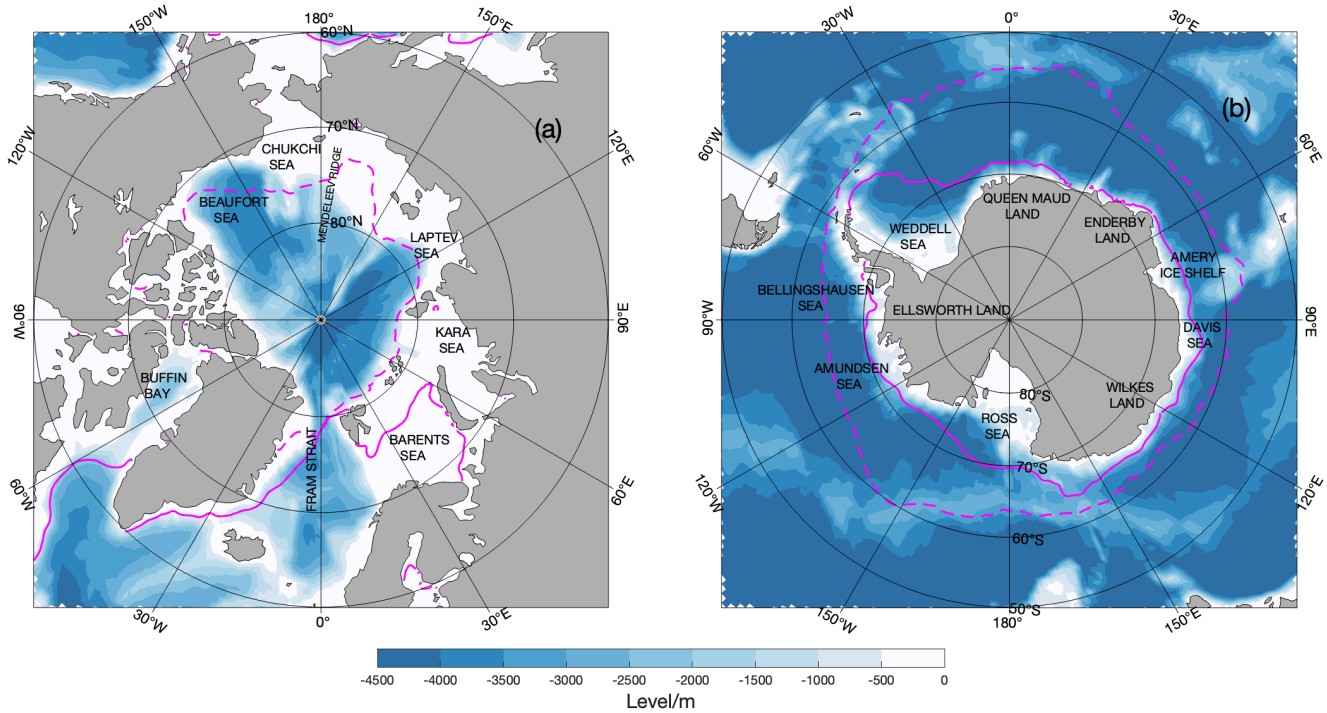


**Figure 1: Study region in (a) Arctic and (b) Southern Oceans. Blue shading represents the bathymetry in meter. Solid and dashed magenta lines indicate the median sea ice extent boundaries for March and September, respectively, defined by areas with sea ice concentration.**

## 2 Data, Method and Processing of the Analysis

### 2.1 Calculation of Ocean-Surface Stress and the Ekman Transport

The ocean-surface stress is estimated using the methodology proposed by Yang (2006, 2009), with modifications by Meneghello et al. (2018). The total ocean-surface stress ($\tau_o$) is calculated as a weighted linear combination of ice–water stress ($\tau_{iw}$) and air–water stress ($\tau_{aw}$), based on the fractional sea ice concentration:

$$\tau_o = \alpha\tau_{iw} + (1-\alpha)\tau_{aw} \tag{1}$$

where $\alpha$ is set to 0 for the ice-free surfaces (defined as sea ice concentration less than 15%) and 1 for ice-covered surfaces (defined as sea ice concentration exceeding 15%). The stresses $\tau_{iw}$ and $\tau_{aw}$ are parameterized using quadratic drag laws:

$$\tau_{iw} = \rho_w C_{D,iw}|U_{ice} - U_e - U_g|(\boldsymbol{U_{ice}} - \boldsymbol{U_e} - \boldsymbol{U_g}) \tag{2}$$

and

$$\tau_{aw} = \rho_a C_{D,aw}|U_{10}|\boldsymbol{U_{10}} \tag{3}$$

where $U_{ice}$, $U_e$, $U_g$, and $U_{10}$ are the local ice motion, Ekman velocity, geostrophic velocity, and equivalent neutral wind at 10-m height, respectively. $\rho_w$ = 1027.5 kg m$^{-3}$ and $\rho_a$ represent the densities of water and air. In this product, $\tau_{aw}$ is taken directly from existing satellite wind products (Yu and Jin 2014a, b).

$C_{D,iw}$ is the ice-water drag coefficient and $C_{D,iw} = 5.5 \times 10^{-3}$ is adopted in this product as it is a commonly recognized value.
It is worth noting that, due to the limited availability of direct observations, $C_{D,iw}$ is identified as a key source of uncertainty.
A sensitivity analysis is therefore provided in the following section to evaluate its potential impact.
In Equation (2), surface ocean velocity expressed as the sum of $U_g$ and $U_e$. The representation of ocean surface stress is known
to be highly sensitive to the assumed surface velocity used in the drag formulation. A range of approaches has been employed
in past studies—incorporating $U_e$, $U_g$, or even assuming zero ocean motion—each with markedly different implications. For
instance, Zhong et al. (2018) showed that mean Ekman pumping in the Beaufort Sea can differ by over 50% depending on the
inclusion of geostrophic flow. Wu et al. (2021) reported similar sensitivities in the Nordic Seas, while earlier works by Zhong
et al. (2015) and Ma et al. (2017) further detailed the variability across Arctic regimes. As a result, stress-based diagnostics
remain sensitive to parameterization choices, and conclusions should be interpreted with that uncertainty in mind.
The geostrophic velocity $U_g$ can be calculated from dynamic ocean topography datasets (McPhee 2013; Armitage et al. 2016,
2017). The Ekman velocity $U_e$, which moves at an angle of 45° to the right of the ocean-surface stress in the Northern
Hemisphere, is calculated as:
$$U_e = \frac{\sqrt{2}e^{-i(\pi/4)}}{f\rho_w D_e}\tau_o \qquad\qquad (4)$$
where $f$ is the Coriolis parameter, and $D_e$ is the Ekman layer depth (20 m, Meneghello et al., 2018). Since $U_e$ and $\tau_o$ are
interdependent in Eqs. (1) and (4), a modified Richardson iteration method is applied to solve them iteratively until converge
is achieved, starting with $U_e = 0$ in the first iteration (Yang 2006).
Subsequently, the vertical Ekman velocity $w_e$ can be calculated as follows:
$$w_e = \frac{1}{f\rho_w}\nabla \times \tau_o \qquad\qquad (5)$$
A positive $w_e$ indicates upwelling, while a negative $w_e$ corresponds to downwelling.

**2.2 Data Description**
The calculation of total ocean-surface stress (Eqs. (1)–(4)) requires the following input datasets: ocean-surface wind stress
($\tau_{aw}$), sea ice concentration ($\alpha$), sea ice motion ($U_{ice}$), and dynamic topography for geostrophic velocity ($U_g$). A brief
description of each satellite-based dataset is given in Table 1.

**Table 1: Gridded satellite datasets used in the work.**

| Variable | Source | Resolution | Period | Reference |
|---|---|---|---|---|
| Surface Wind Stress $\tau_{aw}$ | OAFlux2 | Daily, 0.25° | 1988-present | Yu & Jin, 2014a, 2014b |
| Ice Motion $U_{ice}$ | Polar Pathfinder v4 | Daily, 25 km | 1978-2023 | Tschudi et al., 2019 |

| Geostrophic $U_g$ | multi-altimeter dataset | 3-Day, 25 km | 2011-2021 (Arctic) 2013-2021 (Antarctic) | Prandi et al., 2021 |
|---|---|---|---|---|
| Sea Ice Concentration $\alpha$ | NSIDC0051, v2 | Daily, 25 km | 1988-present | DiGirolamo et al., 2022 |


In this product, the air-water wind stress is taken from OAFlux2 (Yu & Jin, 2014a, 2014b), a satellite-derived 0.25-degree
gridded air-sea flux daily analysis (1988 to present) developed under the auspices of NASA's Making Earth System Data
Records for Use (MEaSUREs) program (Yu, 2019). OAFlux2 winds are synthesized from 19 active and passive satellite wind
sensors and wind stress are calculated from the Coupled Ocean-Atmosphere Response Experiment (COARE) bulk algorithm
version 3.6 (Fairall et al., 2003).
Daily sea ice motion vectors for the Arctic and Antarctic regions are obtained from the National Snow and Ice Data Center's
(NSIDC) Polar Pathfinder Daily 25 km EASE-Grid Sea Ice Motion Vectors, Version 4 (Tschudi et al., 2019, 2020), covering
the period from 1978 through 2023. The ice motion fields are derived from multiple sources, including passive microwave
radiometers (e.g., SSM/I, AMSR-E), visible and infrared sensors (e.g., AVHRR, MODIS), scatterometers (e.g., QuikSCAT),
drifting buoys (e.g., IABP), and atmospheric reanalysis winds. Feature-tracking algorithms are applied to sequential satellite
images to identify ice displacement, while optimal interpolation techniques combine the various data sources to produce daily
sea ice motion estimates. The resulting vectors represent sea ice displacement over a 24-hour period and are gridded onto a 25
km EASE grid 2.0 (EASE2).
Geostrophic velocity in the Arctic and Antarctic are obtained from the CLS/PML multi-altimeter combined Arctic/Antarctic
Ocean sea level dataset (Prandi et al., 2021). This dataset spans latitudes north of 50°N on a 25 km EASE2, with a temporal
resolution of one grid point every 3 days. Covering the Arctic from 2011 to 2021 and the Antarctic from 2013 to 2021, the
CLS dataset mitigates the spurious meridional signals often introduced by the longer sampling intervals of CryoSat-2
observations (Auger et al., 2022).
The Sea Ice Concentrations from Nimbus-7 SMMR and DMSP SSM/I–SSMIS Passive Microwave Data, Version 2 (NSIDC-
0051, Cavalieri et al., 1996; DiGirolamo et al., 2022) is used to define the daily ice boundary based on the 15% ice
concentration threshold. NSIDC-0051 provides a reliable, long-term record of sea ice concentration, making it valuable for
studying sea ice conditions and large-scale climate variability (Parkinson, 2019). Widely recognized for its accuracy, the
dataset is frequently used to validate and improve climate model simulations. The daily dataset is available from 1987 to the
present and provide a coverage on a 25 km resolution polar stereographic grid for the both polar regions.
The analysis period ends in 2018 to maintain consistency with the most reliable iteration of the ongoing refinement of OAFlux
wind product. While this choice limits the temporal extent, the framework itself remains flexible and can be readily extended
as newer, better-resolved datasets become available. With all considered, the study period for Antarctica is constrained to six
years (2013–2018), while an eight-year period (2011–2018) is maintained for the Arctic.
**2.3 Data processing procedure**
Using the methodology described in Eqs. (1)-(5) and the input data listed in Table1, the workflow for processing and analysing
data to calculate ocean-surface stress and derive vertical Ekman velocity is shown in Figure 2.

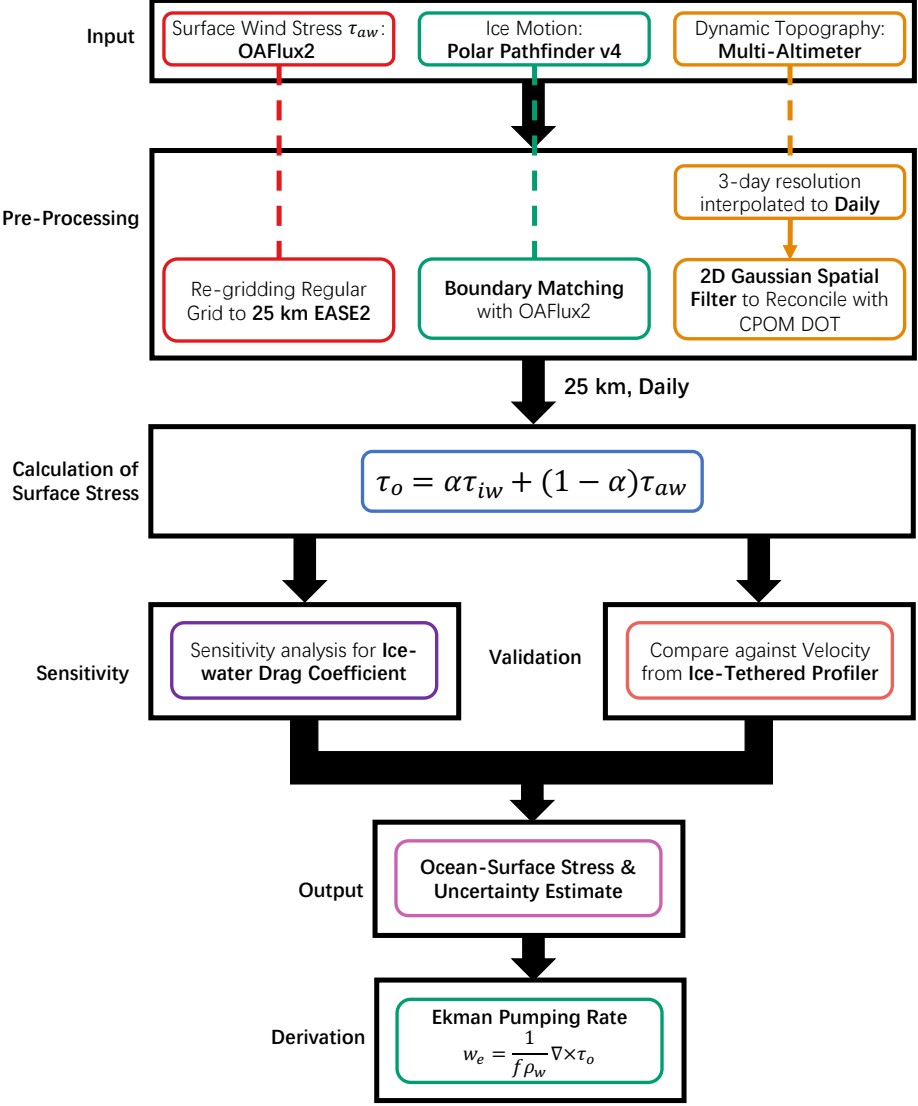


**Figure 2: Workflow for data processing and analysis to calculate ocean-surface stress and derive vertical Ekman velocity.**
All datasets are interpolated onto a common 25 km EASE grid format, providing uniform spatial resolution and facilitating
consistent analysis across the Arctic and Antarctic regions. Although the discrepancies in sea-ice boundaries are very limited
over 2011-2018 (less than 2% of total grids), ice concentration and motion data are adjusted to better align with OAflux2 wind
stress. Noting that the 25 km resolution may introduce uncertainties near the 15% sea ice concentration boundary, as such
coarse resolution can obscure sharp gradients in the marginal ice zone and misclassify mixed ice–water grid cells (e.g., Meier,
2005; Ivanova et al., 2015).
Temporal sampling frequency plays a critical role in determining the accuracy and interpretability of ocean surface stress
estimates. Daily and sub-daily sampling is often needed to capture short-term variability in wind and sea ice motion, which
directly affects transient stress fluctuations and high-frequency Ekman responses (Meneghello et al., 2018; Regan et al., 2020).
Conversely, monthly-averaged fields may overly smooth dynamic features and obscure important stress events, particularly in
regions with strong synoptic variability. In this context, the 3-day sampling of the CLS/PML altimetry product offers a useful
compromise: it resolves large-scale mesoscale dynamics more consistently than monthly data while maintaining better signal-
to-noise properties than noisy daily sea surface height reconstructions in ice-covered regions (Prandi et al., 2021). Given the
limitations of satellite altimetry in the polar oceans, the CLS dataset provides a crucial improvement by reducing spurious
meridional errors and enabling more consistent estimation of geostrophic velocities and their role in modulating surface stress
(Auger et al., 2022). We revisit the implications of time-averaging choices for surface stress/derived velocity fields in Section

175    4.2.

While the CLS/PML product offers improved temporal resolution and geophysical realism at finer spatial scales, its use of
multiple altimeters and interpolation techniques can introduce high-frequency structures that remain difficult to validate given
the sparse in situ coverage at high latitudes (Prandi et al., 2021; Auger et al., 2022). In contrast, the CPOM Dynamic Ocean
Topography (DOT) dataset (2003-2021) from the Centre for Polar Observation and Modelling (CPOM; Armitage et al., 2016,
2017), despite its coarser resolution and monthly cadence, has seen wider adoption and validation across climate-scale Arctic
studies (e.g., Meneghello et al., 2018; Zhong et al., 2018; Lin et al., 2023), making it a valuable benchmark for cross-
comparison. To reconcile the strengths of both datasets, we apply a two-dimensional Gaussian spatial filter to smooth CLS
fields, aligning their effective resolution with CPOM and improving interpretability of large-scale patterns. This hybrid
approach leverages the temporal detail of CLS while benefiting from the broader-scale reliability of CPOM, offering a more
balanced foundation for stress estimation and error characterization in polar oceanographic applications.
We employed a 2D Gaussian filter with a standard deviation of 75 km to improve consistency and interpretability between
CLS and CPOM DOT datasets, which have different resolutions and small-scale characteristics. A sensitivity test is conducted
to determine the optimal filter radius, ranging from 50 km to 250 km. Smaller filters (e.g., <50 km) preserve small-scale
variability but may complicate the interpretation of large-scale features, while larger filters (e.g., >250 km) can excessively
smooth mesoscale processes, such as boundary currents, reducing the dataset's ability to capture key processes of polar
dynamics.
To find the optimal filter size, a series of tests were conducted for 2011. The effectiveness of each filter setting is evaluated
using the Root Mean Squared Deviation (RMSD):
$RMSD = \sqrt{\frac{1}{N}\sum_{i,j}\left(w_{e\,i,j} - w_{e_{ref,(i,j)}}\right)^{2}}$     (6)
where $w_e$ represents the local vertical Ekman velocity $w_e$ derived from the CLS dataset, filtered with a specific Gaussian filter
size (e.g., 100 km, 150 km, etc.), and $w_{ref}$ is the reference vertical Ekman velocity calculated using the CPOM dataset. N is
the total number of the grid points with sea ice coverage.
The unfiltered CLS dataset exhibits clear seasonal variations in RMSD, with values peaking at 25 cm/day during winter and
decreasing in summer (Figure 3a). Applying a Gaussian filter significantly enhances agreement with the CPOM dataset,
reducing RMSD by 10–15 cm/day for most of the year. However, in late summer the reduction is only 2–5 cm/day.
Increasing the filter size further enhances spatial agreement (Figure 3b). From no-filter to a 100 km filter, the annual mean
RMSD is reduced to 17 cm/day, and increasing the filter size to 150 km further lowers the RMSD to 15.5 cm/day. The standard
deviation of daily RMSD is also reduced by half with 150 km filter compared to the unfiltered results. However, larger filter
sizes (e.g. 200 km and 250 km) yield only marginal additional improvements. Therefore, the 150 km Gaussian filter is selected
as a practical and effective balance between preserving spatial features and minimizing small-scale variability for this work.
Figure 3c–h demonstrates the impact of varying filter sizes on the spatial structures of $\tau_o$ and $w_e$ on March 15, 2011. Without
filtering, the CLS dataset exhibits residual meridional striping due to satellite sampling artifacts (Auger et al., 2022). This
pattern is significantly suppressed with a 150 km Gaussian filter. Between the filtered CLS-derived $w_e$ (150 km) and CPOM-
derived $w_e$, the correlation coefficients improving markedly from 0.77 (no filter) to over 0.95 ($p < 0.05$).

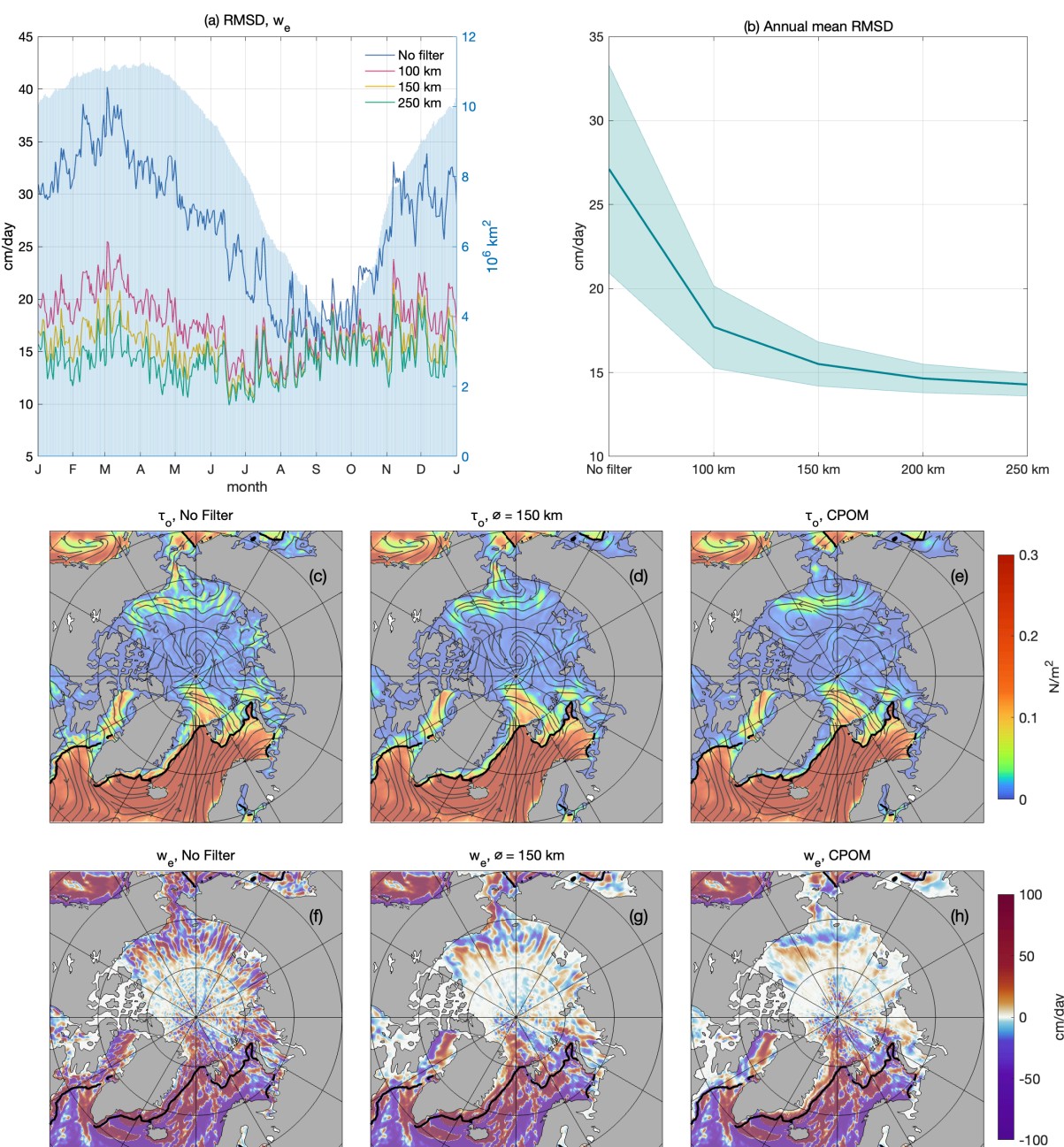


Figure 3: Area-averaged ocean-surface stress $\tau_o$ and vertical Ekman velocity $w_e$ regarding Gaussian filter setting. (a) Annual cycle
of root mean squared deviation (RMSD) of $w_e$ over 2011. Blue shades show total ice-cover areas (right axis). (b) Annual mean RMSD
of $w_e$ with shading indicating one standard deviation over a year. (c) Snapshot of $\tau_o$ with unfiltered CLS (3/15/2011). (d) Same as c
but with 150 km Gaussian filter. (e) Same as c but with CPOM. (f-h) Same as (c-e) but for $w_e$. Streamlines in (c-e) show the direction
of $\tau_o$. Black contours in (c-h) mark the 15% ice concentration on 3/15/2011.

217

## 3 Results and Regional Statistics

### 3.1 Arctic Ocean

In this section, we provide a concise overview of the surface stress and the corresponding Ekman velocity fields. Figure 4a shows the time-averaged ocean-surface stress ($\tau_o$) field across the Arctic for 2011–2018. The highest $\tau_o$ appears in the ice-free Nordic Seas, where strong wind-ocean interactions drive surface stress exceeding 0.3 N/m². In contrast, sea ice reduces momentum transfer and lower the $\tau_o$ in ice-covered regions. In the Seasonal Ice Zone (SIZ), marked by the March and September sea ice boundaries, $\tau_o$ typically remains below 0.05 N/m². Within the Perennial Ice Zone (PIZ), bounded by the September sea ice boundary, it drops further to below 0.02 N/m².

The seasonal cycle of $\tau_o$ is the dominant temporal variability across the Arctic (Stroeve and Notz, 2018). The standard deviation (STD) shows a spatial distribution similar to the time-averaged $\tau_o$ (Figure 4b), with high variability (>0.1 N/m²) in ice-free regions like the Nordic Seas. Variability is significantly suppressed in the SIZ and PIZ, with values below 0.02 N/m² and 0.01 N/m², respectively. The coefficient of determination ($R^2$, here is calculated as the proportion of variance explained by the seasonal cycle, i.e., $R^2 = 1 - \frac{\sum_i (y_i - y_{seasonal})^2}{\sum_i (y_i - \bar{y})^2}$) shows that in open-ocean regions, 40–60% of variance is explained by seasonal variability (Figure 4c). In ice-covered areas, this ratio drops to less than 30%.

The time-mean Ekman pumping rate ($w_e$), alongside its STD and $R^2$ patterns are given in Figure 4d-f. Strong upwelling (>50 cm/day) is observed in the Nordic Seas, while strong downwelling (<-10 cm/day) occurs in the Beaufort and Chukchi Seas. The spatial pattern of STD $w_e$ is similar to that of $\tau_o$ (Figure 4e). Seasonal variability ranges from 10–20 cm/day in ice-free regions, 4–6 cm/day in SIZ, and falls below 4 cm/day in the PIZ. Seasonal variability accounts for up to 60% of $w_e$ variance south of the Denmark Strait, but in other regions, including both ice-covered and ice-free zones, it typically explains 10–30%.

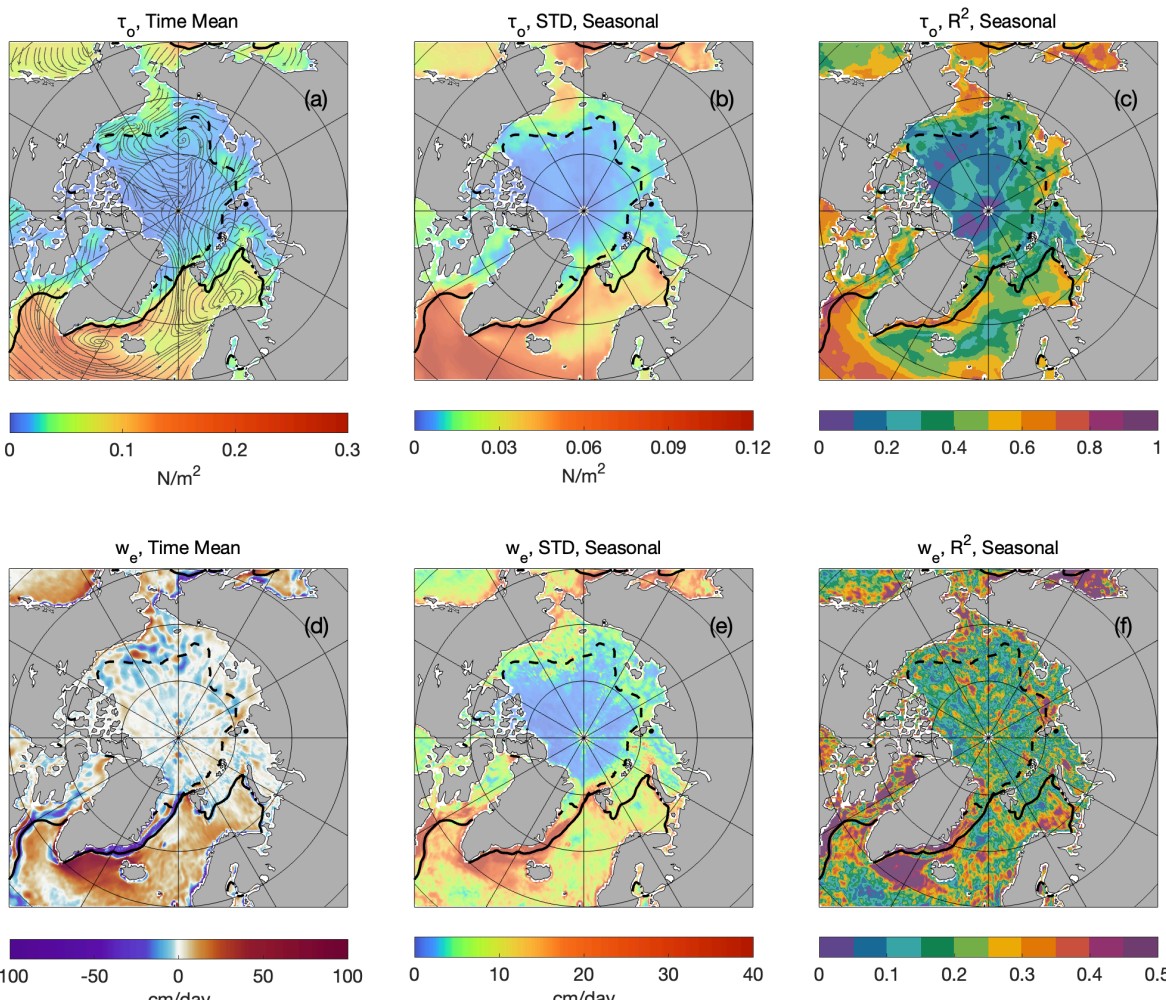

Figure 4: Mean and variability of ocean-surface stress $\tau_o$ and Ekman pumping rate $w_e$ (positive indicates upwelling, negative indicates downwelling) in the Arctic region over 2011-2018. (a) Mean $\tau_o$, with streamlines indicating the direction of stress. (b) Standard deviation of $\tau_o$ seasonal variability. (c) $R^2$, representing $\tau_o$ variance explained by seasonal variability. (d-f) Same as (a-c) but for $w_e$. Streamlines in (a) show the direction of $\tau_o$. The solid and dashed black lines represent the March and September sea ice boundaries, respectively, defined by 15% sea ice concentration averaged over 2011-2018.

The seasonal cycle of area-averaged wind-ocean surface stress ($\tau_{aw}$) is marked by strong values in winter, peaking around 0.4 N/m² and much weaker values in summer, dropping below 0.05 N/m² (Figure 5a). This variation corresponds to the seasonal retreat of sea ice and the associated expansion of open ocean during summer months.

In ice-covered regions, the seasonal cycle of ice-ocean surface stress ($\tau_{iw}$) is similar to that of $\tau_{aw}$, though with significantly lower magnitudes (Figure 5c). The seasonal peak of $\tau_{iw}$ is slightly higher in 2018 than 2013, increasing from 0.010 N/m² to nearly 0.018 N/m².

In ice-free regions, the average pumping rate $w_{e,aw}$ peaks during winter upwelling, reaching around 30 cm/day, and transitions
to weak downwelling during the summer (Figure 5e). Annual variation in winter maximum upwelling rate is evident, with a
notable decline to 10 cm/day by late 2018 (Figure 5f). In contrast, in ice-covered regions, $w_{e,iw}$ is predominantly negative
(Figure 5g), although occasional summer upwelling events occur on daily scales. Notably, the winter downwelling rate has
decreased from approximately -8 cm/day in 2013–2014 to about -4 cm/day by 2017.

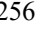



## 3.2 Southern Ocean

The spatial distribution of the time-mean $\tau_o$ in the Antarctic region is shown in Figure 6a. $\tau_o$ exhibits a prominent circumpolar
pattern. In ice-free regions, $\tau_o$ typically ranges from 0.2 to 0.3 N/m². In the SIZ, $\tau_o$ decreases significantly, falling to 0.04–
0.06 N/m², with strong regional variability.
The STD of $\tau_o$ seasonal variability is evidently strong near the September sea ice boundary, exceeding 0.1 N/m², particularly
between 0-90°E (Figure 6b). Moving northward into subpolar open-ocean, the STD gradually declines to approximately 0.04
N/m². Within the SIZ, seasonal variability diminishes further, typically ranging from 0.02 to 0.04 N/m². In the PIZ, it drops
below 0.02 N/m². The R² shows that in regions such as the Indian Ocean and southeast Pacific, seasonality explains over 50%
of the total variance, while in other areas, this proportion ranges from 20% to 40% (Figure 6c).
The spatial structure of the time-mean $w_e$ reveals widespread upwelling south of 50°S (Figure 6d), extending nearly all the
way to the coast of Antarctica. In contrast to the Arctic, where strong ice-ocean coupling leads to clear transitions between
upwelling and downwelling across ice boundaries, the Southern Ocean does not exhibit this distinct pattern. Downwelling is
generally found around 55°S and farther north, or more narrowly along the Antarctic coastline.
The STD pattern of seasonal variability in $w_e$ is relatively consistent across the Southern Ocean (Figure 6e), regardless of sea-
ice coverage, with an average value of approximately 10 cm/day. Higher variability, reaching up to 20 cm/day, occurs only
near the September ice boundary and is very localized. The R² pattern is also relatively homogeneous, with most areas showing
seasonal variability accounting for about 30% of the variance. Along the east coast of Antarctica, the seasonal cycle explains
more than 50% of the variance.

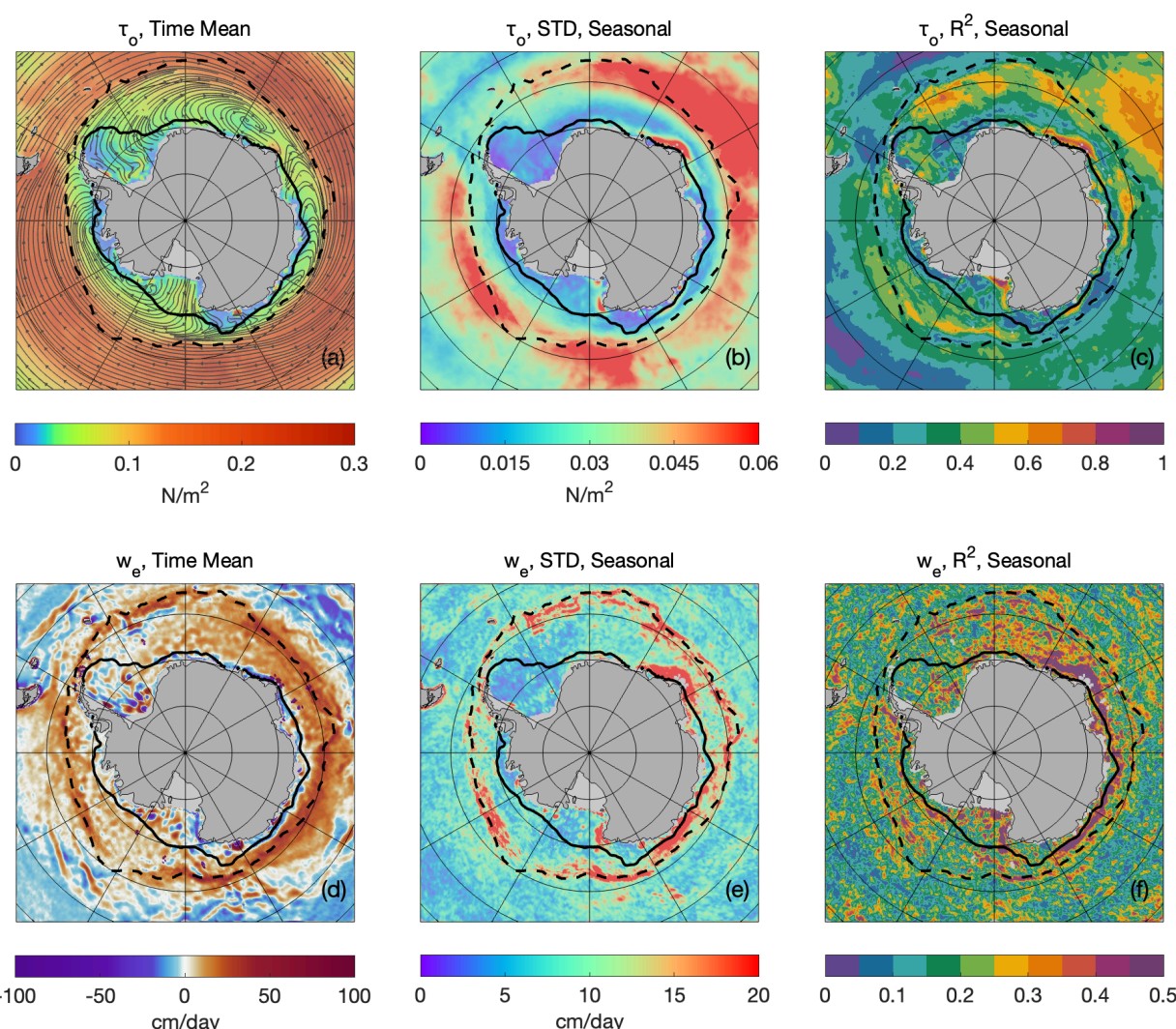


**Figure 6: Mean and variability of ocean-surface stress $\tau_o$ and Ekman pumping rate $w_e$ (positive indicates upwelling, negative indicates downwelling) in the Antarctic region over 2013-2018. (a) Mean $\tau_o$, with streamlines indicating the direction of stress. (b) Standard deviation of $\tau_o$ seasonal variability. (c) $R^2$, representing $\tau_o$ variance explained by seasonal variability. (d-f) Same as (a-c) but for $w_e$. Streamlines in (a) show the direction of $\tau_o$. The solid and dashed black lines represent the March and September sea ice boundaries, respectively, defined by 15% sea ice concentration averaged over 2013-2018.**

290

The seasonal cycle and time series of area-averaged air-water stress $\tau_{aw}$ in the Antarctic are shown in Figures 7a and 7b. In ice-free regions of the Antarctic, the average $\tau_{aw}$ peaks in August at 0.36 N/m² and reaches its minimum in January at 0.13 N/m². Annual variability is relatively small, ranging between 0.022 and 0.026 N/m², with a notable positive anomaly in 2015, when the annual mean briefly increased to 0.028 N/m².

In ice-covered regions, ice-water stress $\tau_{iw}$ shows a delayed seasonal cycle compared to $\tau_{aw}$ and peaks in September (Figure 7c). It is approximately one-fifth to one-half of $\tau_{aw}$, ranging between 0.02 N/m² and 0.08 N/m². The seasonal pattern is

asymmetrical and aligns with the seasonal cycle of sea ice coverage (Eayrs et al., 2019). Similar to the Arctic, the area-averaged summer minima of $\tau_{iw}$ is slightly higher in 2018 compared to 2013, increasing from 0.010 N/m² in to 0.022 N/m².

The seasonal cycle of open-ocean Ekman pumping rate $w_{e,aw}$ is relatively weak (Figure 7e), with higher values in winter (12 cm/day) and lower values in summer (5 cm/day). The absence of a distinct seasonal signal is likely due to the weaker seasonal cycle observed in 2017 and 2018 (Figure 7f). The annual mean varies narrowly between 7 and 9 cm/day.

In ice-covered regions, $w_{e,iw}$ is mostly positive throughout the year, with a brief downwelling period between January and April. A shift toward stronger downwelling occurs in February, with mean values decreasing from -2 cm/day in 2013 to nearly -10 cm/day by 2017. A notable anomaly occurred in 2015 when the annual mean rose sharply from 2 cm/day to 4 cm/day.

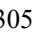

**Figure 7: Timeseries and seasonal cycle of area-averaged surface stress $\tau_o$ (red) and Ekman pumping rate $w_e$ (orange, positive indicates upwelling, negative indicates downwelling) for the Antarctic region. Total areas $a$ of the corresponding areal coverage are also plotted in blue. Variables are subscripted $aw$ when averaged/summed over ice-free open ocean, and are subscripted $iw$ when averaged/summed over ice-covered open ocean. Annual and monthly means are shown as dots in all panels. (a) Seasonal cycle of $\tau_{aw}$ over the ice-free open ocean. (b) Timeseries of $\tau_{aw}$ from 2013-2018. (c) Seasonal cycle of $\tau_{iw}$ over the ice-covered ocean. (d) Timeseries of $\tau_{iw}$ from 2013-2018. (e-h) same as (a-d) but for $w_e$.**

## 4 Uncertainty and Data Quality Assessment

### 4.1 Sensitivity Analysis of Ice-Water Drag Coefficient and uncertainty estimate

The ice-water drag coefficient, $C_{D,iw}$ is often assumed to be constant across time and space due to the scarcity of direct observations that capture its spatiotemporal variability. However, $C_{D,iw}$ can vary significantly depending on environmental conditions such as wind and wave dynamics, ice roughness, sea ice concentration, and surface morphology (Lüpkes et al., 2012; Lüpkes and Birnbaum, 2005; Cole & Stadler, 2019). Reported values for $C_{D,iw}$ range from 0.7 to over $10.0 \times 10^{-3}$ (Overland, 1985; Guest and Davidson, 1987, 1991; McPhee, 2008; Cole et al., 2014), with some extreme cases reaching magnitudes on the order of $10.0 \times 10^{-1}$ (Kawaguchi et al., 2024).

Commonly, a representative value of $5.5 \times 10^{-3}$ has been widely adopted as a pragmatic approximation by the scientific community (Guest and Davidson, 1987; Anderson, 1987). However, this approximation may overlook important spatial and temporal variations in $C_{D,iw}$, highlighting the need for ongoing efforts to improve observations and refine its parameterization. To evaluate the sensitivity of estimated $\tau_o$ to the variations in $C_{D,iw}$, two sets of experiments are conducted for 2011: one with fixed $C_{D,iw}$ values ranging from $1.0 \times 10^{-3}$ to $10.0 \times 10^{-3}$, and another using a randomized weighting map, dynamically varying $C_{D,iw}$ between between order of $10^{-3}$ and $10^{-2}$ on a daily basis at each grid cell.

The amplitude of $\tau_o$ scales proportionally with $C_{D,iw}$, as implied from Eq. 2 (Figure 8a). For fixed coefficients, the summer mean $\tau_o$ increases from 0.003 N/m² at $C_{D,iw} = 1.0 \times 10^{-3}$, to 0.015 N/m² at $C_{D,iw} = 10.0 \times 10^{-3}$, while winter means rise from 0.012 N/m² to 0.053 N/m². Results from the random-weighted $C_{D,iw}$ experiment closely follows the fixed cases of $C_{D,iw} = 5.0$-$6.0 \times 10^{-3}$. Similarly, the annual mean $\tau_o$ and its standard deviation increase proportionally with $C_{D,iw}$ (Figure 8b), quadrupling the annual mean and raising the standard deviation from 0.003 N/m² to 0.017 N/m² as $C_{D,iw}$ increases.

Figures 8c–h show the spatial distribution of $\tau_o$ and $w_e$ in response to varying $C_{D,iw}$. Under low $C_{D,iw}$ circumstances, momentum transfer between ice and ocean is reduced, leaving small scale variability indistinct particularly in the central Arctic. As $C_{D,iw}$ increases, regions with high surface stress intensify, particularly in areas like Baffin Bay, the Chukchi Sea, and north of Fram Strait.

At $C_{D,iw} = 1.0 \times 10^{-3}$, the Ekman pumping rate in regions like the Fram Strait barely reach ±8 cm/day, whereas at $C_{D,iw} = 10.0 \times 10^{-3}$, it exceeds ±30 cm/day, with strong contrasting upwelling and downwelling patterns. additionally, while the random-

weighted $C_{D,iw}$ experiment introduces spatial noise, the broader spatial structures of both $\tau_o$ and $w_e$ remain consistent with
fixed-coefficient runs.

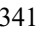

**Figure 8: Area-averaged ocean-surface stress $\tau_o$ and Ekman pumping rate $w_e$ regarding different $C_{D,iw}$. (a) annual cycle of $\tau_o$ of**
**2011, area-averaged over the sea ice-cover region. Blue areas show total ice-cover areas (right axis); (b) annual mean of $\tau_o$ with**
**shading indicating one standard deviation over a year. Red dashed line marks $C_{D,iw} = 5.5 \times 10^{-3}$, black dotted line shows the annual**
**mean of random $C_{D,iw}$ experiment. (c) Snapshot of $\tau_o$ with $C_{D,iw} = 1.0 \times 10^{-3}$ (3/15/2011). (d) Same as c but with $C_{D,iw} = 10.0 \times 10^{-3}$.**
**(e) Same as c but with random $C_{D,iw}$. (f-h) Same as (c-e) but for $w_e$. Streamlines in (c-e) show the direction of $\tau_o$. Black contours in**
**(c-h) mark the 15% ice concentration on 3/15/2011.**
The final estimated uncertainty $\varepsilon_{iw}$, in the ice-water stress $\tau_{iw}$ is quantified daily through the integration of standard errors
from sensitivity analyses of $C_{D,iw}$ and spatial Gaussian filter tests. Both filter tests and $C_{D,iw}$ tests are extended to the full
analysis period: eight years (2011–2018) for the Arctic and six years (2013–2018) for the Antarctic. Using the root-sum-square
method, the combined uncertainty is expressed as:
$$\varepsilon_{iw} = \sqrt{(\varepsilon_{iw,F})^2 + (\varepsilon_{iw,C})^2} = \sqrt{(\frac{\sigma_F}{\sqrt{N_1}})^2 + (\frac{\sigma_C}{\sqrt{N_2}})^2} \qquad (7)$$
where $\sigma_F$ is the standard deviation of $\tau_{iw}$ from different Gaussian filter settings, and $\sigma_C$ represents the standard deviation of
$\tau_{iw}$ from sensitivity analysis on varying $C_{D,iw}$. The terms $N_1$ and $N_2$ denote the number of runs performed in each sensitivity
analysis. This estimate assumes independence between $C_{D,iw}$ and geostrophic fields (which were spatially filtered), with
perturbations of comparable amplitude between the two sets of sensitivity analysis.
Figure 9 shows the spatial distributions of relative uncertainty ($\varepsilon_{iw}$ to $\tau_{iw}$) for the Arctic (March 15, 2013) and the Southern
Ocean (September 15, 2013) during winter season. Overall, spatial filtering produces scattered patterns (Figures 9a–9d), while
varying ice-water drag coefficient yield smoother distributions (Figures 9e–9h). Median uncertainties are comparable between
the two sets of experiments, ranging from 14–18% in the Arctic to 22–25% in the Antarctic. The greater uncertainties in the
Antarctic reflect higher local stress variability and increased sensitivity to parameter changes, which also manifests as the
higher uncertainties observed in winter compared to summer (Figures 3 and 8).
In the Arctic, combined uncertainties for zonal surface stress ($\tau_x$) typically range from 10–20%, while locally it could exceed
100% along dynamic regions such as the Fram Strait and Beaufort Sea. Meridional stress ($\tau_y$) exhibits similar spatial
distributions, but with higher uncertainties near the Mendeleev Ridge. Median uncertainty levels for both zonal and meridional
components are below 20%.
Conversely, Antarctic uncertainties are substantially higher, with median values around 40%. The highest uncertainties (>60%)
are concentrated near the sea ice boundary, particularly in the eastern Weddell and Ross Seas. Regional hotspots include the
Antarctic Peninsula and west of Ross Sea for $\tau_x$, and Enderby Land and the Amundsen Sea for $\tau_y$.

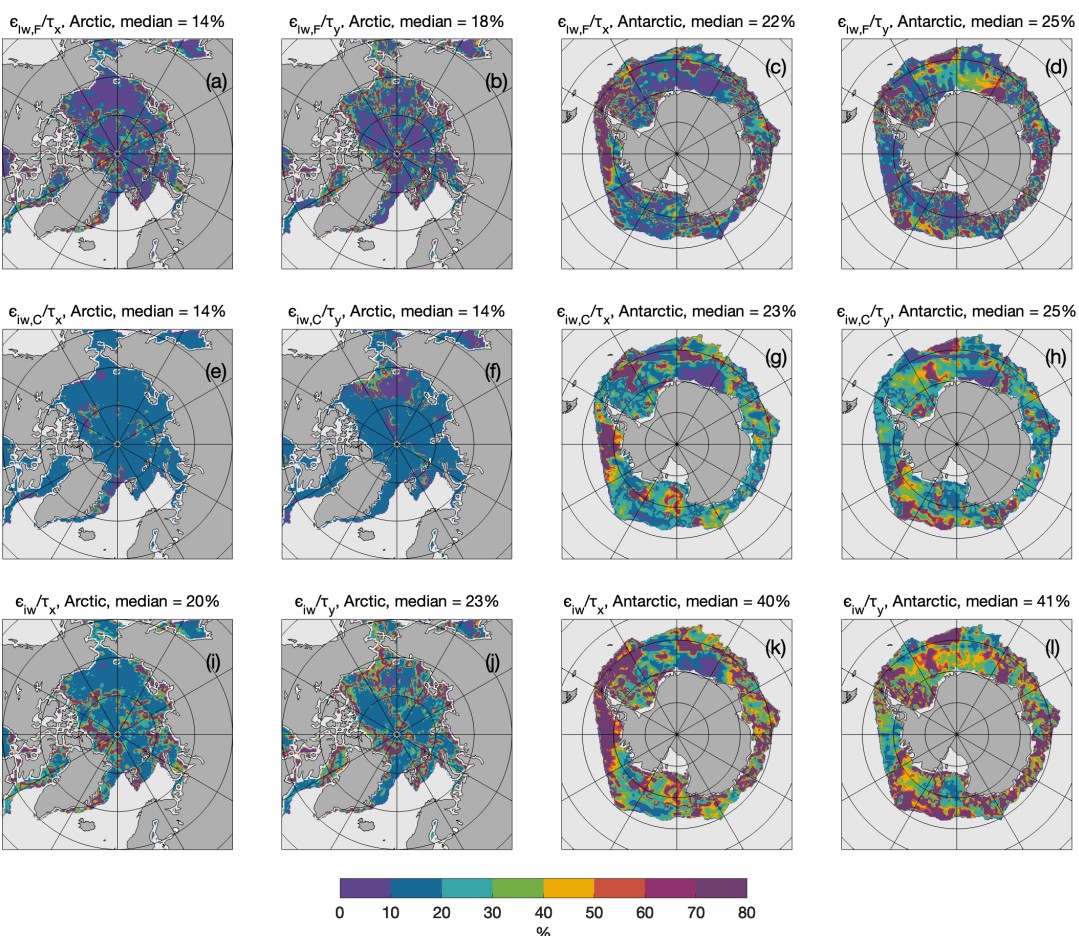


**Figure 9: Estimated uncertainty fields for zonal and meridional ice-water surface stress, expressed as a ratio to the estimated ice-water surface stress in the Arctic (3/15/2013) and Southern Ocean (9/15/2013). (a) standard error introduced by Gaussian filter in the Arctic, zonal direction; (b) error from filter in the Arctic, meridional direction; (c-d) same as (a-b) but for the Antarctic; (e-h) same as (a-d) but for standard error in ice-water drag coefficient $C_{D,iw}$; (i-l) same as (a-d) but for the combined uncertainty.**


In addition to the sensitivity analyses presented for drag coefficient magnitudes and spatial filtering, several other sources of
uncertainty may influence the accuracy of the derived surface stress fields. First, uncertainties in the atmospheric reanalysis
products used as forcing data—particularly in wind speed and direction over sea ice—can propagate directly into surface stress
estimates. Prior studies have shown that different ice motion products can yield differences of at least 20–30% in polar regions
due to discrepancies in boundary-layer representation, data assimilation techniques, and satellite retrieval biases (Sumata et
al., 2014; Wang et al., 2022). These differences are especially pronounced in the marginal ice zone, where sharp gradients in
atmospheric and surface properties are common (Wang et al., 2021; Boutin et al., 2020).

Second, the role of ocean and atmospheric stratification is not explicitly resolved in our parameterization, yet it can
significantly affect stress transmission through the ice–ocean interface. Observational and modeling studies (Lüpkes &
Gryanik, 2015; Lüpkes et al., 2012; Brenner et al., 2021) have shown that stability conditions in the atmospheric boundary
layer modulate drag coefficients by altering turbulence and momentum fluxes—especially under stable stratification, common
in winter Arctic conditions. Likewise, vertical stratification in the upper ocean can modify Ekman layer dynamics and the
effective depth over which stress-induced velocities operate, introducing further uncertainty in estimates of vertical transport
(Meneghello et al., 2018; Zhong et al., 2018).
Third, the spatial and temporal scales over which stresses are calculated can introduce methodological uncertainty. Coarse
averaging may obscure high-frequency processes such as synoptic wind events, inertial motions, or mesoscale eddies, while
finer-scale estimates risk amplifying local noise or aliasing undersampled variability (Timmermans et al., 2008; Manucharyan
& Thompson, 2017; Alberello et al., 2020). This is particularly relevant in the marginal ice zone, where surface properties
evolve rapidly. A more detailed analysis of scale effects and filtering sensitivity is presented in the following section.
Together, these factors point to the need for caution when interpreting surface stress magnitudes or derived quantities like
Ekman pumping, particularly when used to constrain physical budgets or force ocean models. Future work should prioritize
uncertainty quantification through ensemble reanalysis comparisons, the inclusion of stratification effects in drag
parameterizations, and adaptive filtering techniques that respond to local dynamical conditions.

## 4.2 Validation with ITP Observations

**4.2 Validation with ITP Observations**
Since surface stress is not usually directly measured, assessing the performance of our analysis is challenging. To address this,
we revisit the assumption that surface velocity comprises both Ekman and geostrophic components, as described in Eq. (2).
The geostrophic velocity ($U_g$) is derived from the dataset provided by Prandi et al. (2021), while the Ekman velocity component
($U_e$) can be easily calculated from the ocean-surface stress using Eq. (4).
This assumption provides a first-order approximation of surface velocity, and neglects other processes, such as ageostrophic
motions, vertical shear, and submesoscale dynamics, which may introduce additional uncertainties. To robustly validate
satellite-derived ocean surface stress estimates, we compared the derived surface velocity, i.e., sum of $U_g$ and $U_e$, with in situ
measurements from Ice-Tethered Profilers (ITPs, Krishfield et al., 2008; Toole et al., 2011; http://www.whoi.edu/itp). In
particular, several ITPs that equipped velocity sensors (ITP-V, Williams et al., 2010) are used. To align the temporal resolution
of the datasets, we processed the ITP data by computing daily and weekly means, facilitating direct comparisons with daily
satellite products. This approach avoids the uncertainties associated with interpolating satellite data to match the higher-
frequency ITP profiles, which could introduce significant errors due to the under-sampling nature of satellite observations.
Despite this temporal alignment, inherent limitations persist due to spatial and temporal sampling discrepancies. ITPs provide
high-resolution vertical profiles at specific locations, capturing fine-scale and transient oceanic features. In contrast, satellite
observations offer broader spatial coverage but may not resolve such fine-scale variability, especially in polar regions where
data gaps are common due to persistent cloud cover and sea ice. These differences can lead to reduced correlation and increased
bias in validation statistics, as observed in previous studies comparing satellite-derived sea surface salinity products with in
situ observations (Thouvenin-Masson et al., 2022; Boutin et al., 2016; Vinogradova & Ponte, 2013).
It is important to note that this comparison does not serve as a definitive validation of the absolute accuracy of our stress
estimates. Instead, it assesses whether the foundational assumptions underpinning our analysis sufficiently represent the
complex dynamics of the Arctic Ocean.
We use velocity data collected from five ITP-V missions deployed on multiyear sea ice in the Canada Basin between 2011
and 2019 (Figure 10; Table 2). Observations from ITP-77, ITP-78, and ITP-79 are truncated to exclude periods with significant
data gaps and drifts near the end of their deployments.
The five ITP-Vs are categorized into two groups based on deployment timing and drift trajectories. ITP-70 and ITP-80 were
deployed during summer, operated for ~300 days, and primarily drifted between 75–80°N. In contrast, ITP-77, ITP-78, and
ITP-79, deployed in March 2014, operated for less than 200 days and followed more constrained east-to-west trajectories
between 73–75°N.

**Table 2: Details of the ITP-V records**

| Unit ID | Start | | Last | | # of Days | # of Profiles |
|---------|-------|------|------|------|-----------|---------------|
| | Position | Date | Position | Date | | |
| ITP-70 | 76.81°N 138.89°W | 8/26/2013 | 77.11°N 156.51°W | 7/15/2014 | 324 | 3713 |
| ITP-77 | 73.37°N 134.99°W | 3/11/2014 | 75.89°N 158.50°W | 10/2/2014 | 206 (153[*]) | 2367 (1800[*]) |
| ITP-78 | 74.36°N 135.14°W | 3/12/2014 | 74.08°N 145.43°W | 8/6/2014 | 148 (130[*]) | 1694 (1500[*]) |
| ITP-79 | 75.38°N 136.50°W | 3/22/2014 | 75.02°N 148.37°W | 9/30/2014 | 193 (143[*]) | 1694 (1636[*]) |
| ITP-80 | 77.36°N 146.15°W | 8/14/2014 | 75.68°N 151.79°W | 5/24/2015 | 284 | 3260 |

[*] Data towards the end of the series exhibits quality issues that necessitate truncation.

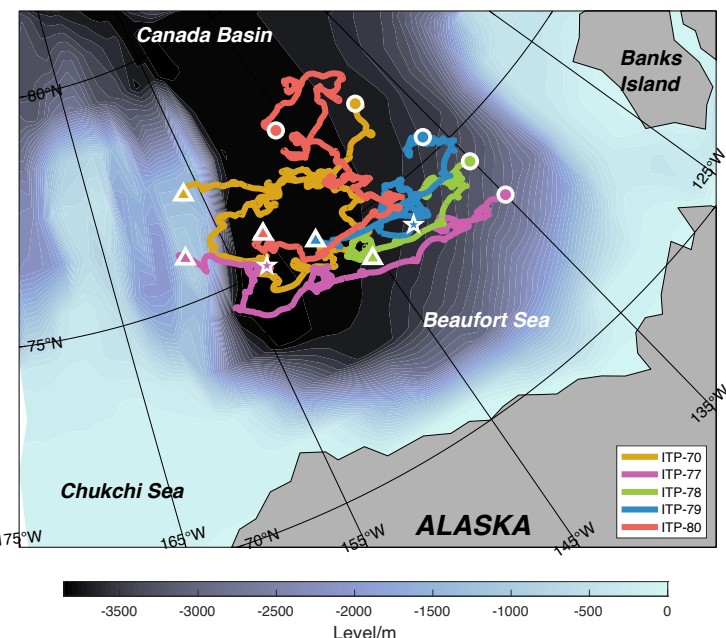


**Figure 10: ITP-V drift paths in the Arctic Ocean (colored curves). The deployment locations are marked by circles, latest locations**
**by triangles, and the cutoff locations for ITP-77 and ITP-79 by stars.**


To account for temporal sampling differences and mitigate aliasing from high-frequency variability, the sub-daily ITP-V
velocity data are first averaged to daily means before comparison with the satellite-derived velocity field ($U_g + U_e$). The
corresponding satellite values are then extracted at the nearest grid point along each ITP track (Figure 11). This approach
reduces mismatch due to subsampling in the satellite product and ensures a more consistent temporal basis for comparison.
Along the path of ITP-70 and ITP-80, satellite-derived ocean surface velocities exhibit moderate agreement with in situ
observations, particularly in capturing high-frequency variability. In contrast, comparisons with ITP-77, ITP-78, and ITP-79
reveal weaker correspondence, most notably in the zonal velocity components. For the meridional component, ITP-77 shows
relatively better alignment during the initial ~100-day period until mid-July.

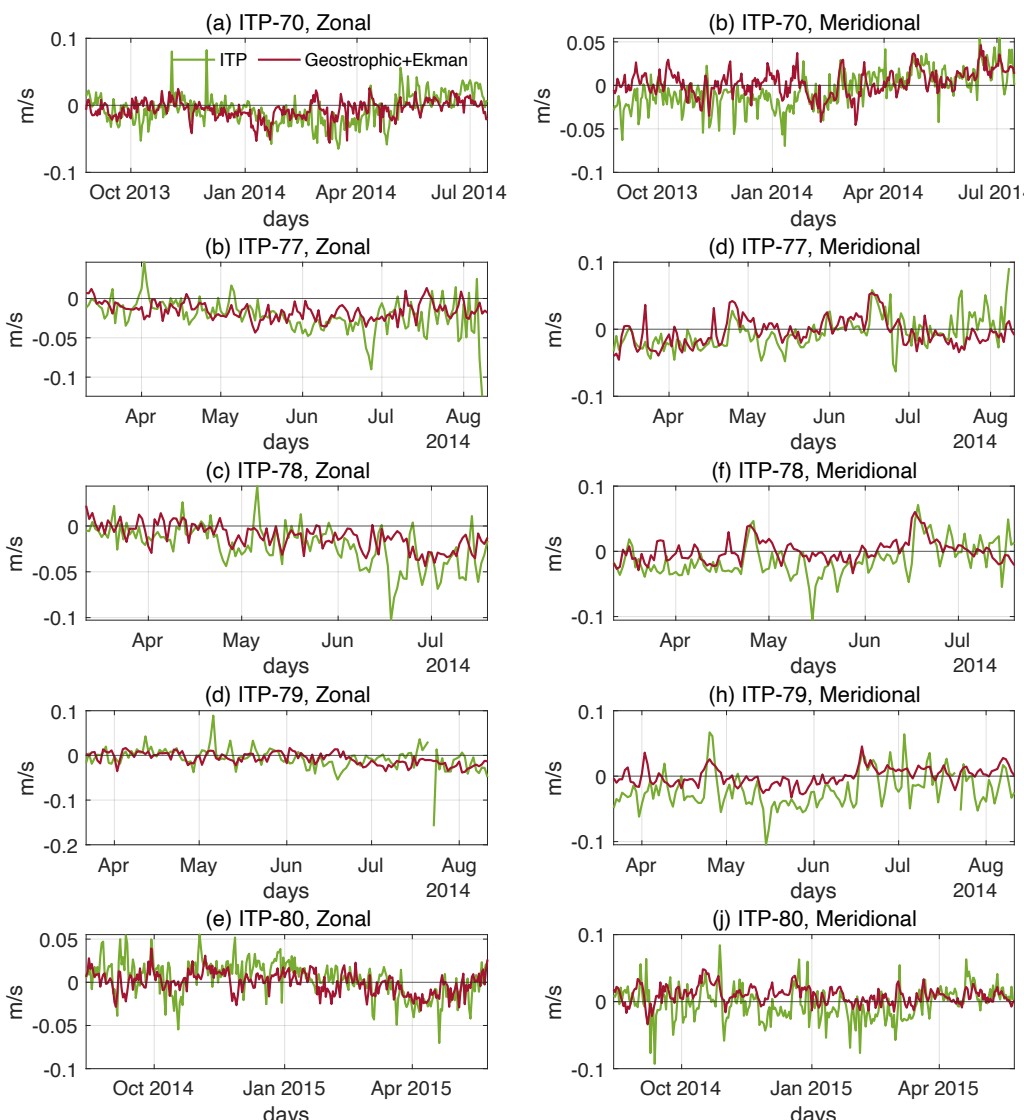


**Figure 11: Daily mean timeseries of zonal and meridional surface velocity. (a) Zonal velocity along ITP-70 paths. (b) Zonal velocity along ITP-77 paths. (c) Zonal velocity along ITP-78 paths. (d) Zonal velocity along ITP-79 paths. (e) Zonal velocity along ITP-80 paths. (f-j) same as (a-e) but for meridional velocity. Green curves represent velocity data retrieved from ITPs. Red curves are collocated obtained from the satellite-derived velocity fields, i.e., geostrophic plus Ekman velocity.**


Figure 12 presents the comparison of satellite-derived surface velocity components against collocated ITP-V observations for
the ITP-70 (panels a,b) and ITP-80 (panels c,d) paths. For ITP-70, the zonal component yields a Pearson correlation of $r =$
0.31 and standard deviation of 0.022 m s$^{-1}$, while the meridional component gives $r = 0.42$ and STD = 0.020 m s$^{-1}$. ITP-80
exhibits slightly stronger zonal agreement ($r = 0.43$) but weaker meridional agreement ($r = 0.34$). In both deployments, scatter
markedly decreases for observations taken after ~200 days (warm colors), accompanied with the predominantly northward
drift of ITP-70 and westward drift of ITP-80 seen in Figure 10.

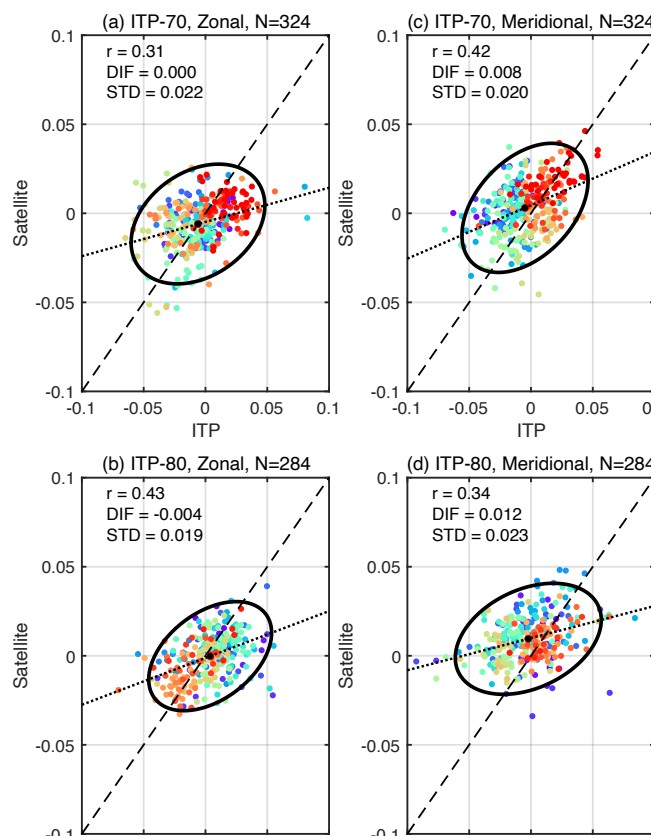

**Figure 12: Scatterplots of collocated surface velocity pairs for ITPs with data spanning more than 200 days (unit: m/s). (a) Zonal**
**velocity along ITP-70 paths. (b) Zonal velocity along ITP-80 paths. (c-d) Same as (a-b) but for meridional velocity. The total number**
**of days (N) is given. Correlation coefficients, mean differences (DIF), and standard deviations (STD) of the differences between**
**satellite-derived velocity and ITP observations are also displayed. 95% confidence ellipse (black contour), linear fitting (black dotted**
**line) are also given in each panel.**

By contrast, Figure 13 summarizes the additional ITP observational periods (panels a–f), where correlation coefficients span
$r = 0.07\text{–}0.56$, and STD = 0.021–0.025 m s$^{-1}$. One component reaches a moderate correlation ($r = 0.56$), while most remain
weak ($r < 0.40$), and no coherent temporal clustering is apparent. Across all deployments, satellite-derived velocities exhibit a
slight northward bias and tend to underestimate ITP-measured surface speeds beyond 100 days post-release.

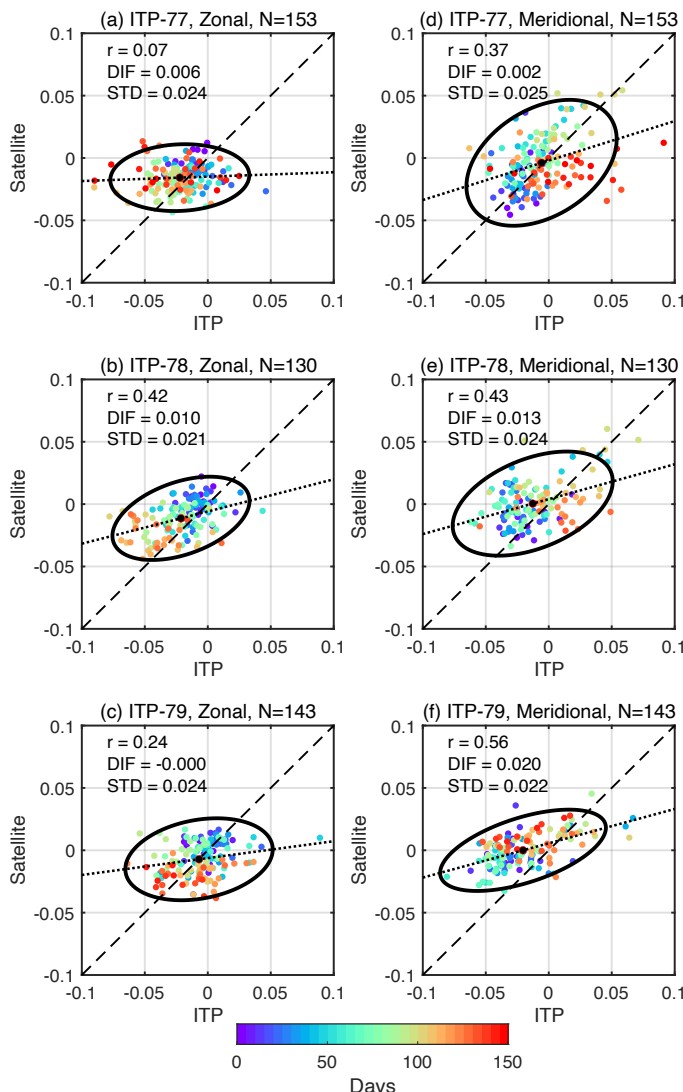


**Figure 13: Scatterplots of collocated surface velocity pairs for ITPs with data spanning less than 200 days (unit: m/s). (a) Zonal velocity along ITP-77 paths. (b) Zonal velocity along ITP-78 paths. (c) Zonal velocity along ITP-79 paths. (d-f) Same as (a-c) but for meridional velocity. The total number of days (N) is given. Correlation coefficients, mean differences (DIF), and standard deviations (STD) of the differences between satellite-derived velocity and ITP observations are displayed. 95% confidence ellipse (black contour), linear fitting (black dotted line) are also given in each panel.**

480

Table 2 and Figure 14 provide a comprehensive comparison between satellite-derived surface velocity estimates and in situ velocity measurements obtained from ITP-V across five deployments. The analysis includes both the zonal (east–west) and meridional (north–south) velocity components and considers statistics derived from both daily and weekly averaged timeseries. A consistent northward bias is evident in the satellite-derived velocities across most ITP paths. While the mean zonal velocities are generally in close agreement between satellite and ITP-V data—for instance, at ITP-70 both sources report a mean of –

0.006 m/s—some deployments exhibit a more pronounced bias. ITP-77 and ITP-78, in particular, show a noticeable eastward
offset, with satellite-derived zonal velocities being more positive than the ITP-V counterparts by approximately 0.005–0.01
m/s. The meridional component aligns more closely overall, but satellite velocities still tend to be more northward. These
biases are evident in the top-left panel of Figure 14, where most points lie above the 1:1 line, especially in the meridional
direction.
In addition to the bias, satellite-derived velocities systematically exhibit reduced variability compared to ITP-V observations.
Across all deployments and components, the STD of satellite velocity is consistently lower than that observed in the ITP-V
data. For example, while the average zonal STD in the ITP data is around 0.022 m/s, the corresponding satellite value is
approximately 0.011 m/s. The top-right panel of Figure 14 illustrates this discrepancy clearly: all data points fall below the 1:1
line, indicating that satellite products underestimate temporal variability. This reduced variability likely reflects the filtering
and smoothing inherent in satellite altimetry products, which are designed to represent large-scale geostrophic flows and may
not fully resolve the higher-frequency or smaller-scale fluctuations captured by the ITP instruments.
Despite these limitations, satellite-derived velocity fields are able to explain a substantial portion of the observed variance in
the ITP-V measurements on daily mean scale. The coefficient of determination ($R^2$) values indicate that, on average, satellite
products account for about 50% of the variability in the daily data. This explanatory power increases substantially when the
analysis is performed on weekly averaged time series, with $R^2$ reaching as high as 0.77 for the zonal component at ITP-78 and
0.60 for the meridional component at ITP-79. These results suggest that, although satellite estimates smooth out finer-scale
variability, they capture the dominant patterns in large-scale motion effectively. However, the correlation coefficients between
satellite and ITP-V velocities are more modest, typically ranging from 0.3 to 0.4 in the daily records. With weekly averaging,
these correlations improve significantly, sometimes by more than 0.2, as illustrated in the bottom-left panel of Figure 14. This
panel shows weekly-averaged points clustering nearer to or above the diagonal, particularly for ITP-78 and ITP-79, while daily
correlations tend to remain lower and more scattered.
The performance of the satellite velocity estimates varies by deployment. ITP-78 and ITP-79 demonstrate the strongest
agreement. For example, ITP-78's zonal component yields correlation coefficients of 0.42 for daily data and 0.78 for weekly
data, with $R^2$ values of 0.32 and 0.77, respectively. Similarly, ITP-79's meridional component shows a daily correlation of
0.56 and a weekly correlation of 0.76, with corresponding $R^2$ values of 0.47 and 0.60. These high values underscore the ability
of satellite altimetry to capture meaningful geophysical signals under favorable conditions. Conversely, performance is notably
weaker at ITP-77, where the zonal velocity component yields a daily correlation of only 0.07 and a weekly correlation of 0.31,
suggesting a diminished ability of satellite products to resolve local variability in that particular region or time frame. Such
differences likely arise from a combination of regional oceanographic complexity and satellite data limitations, including issues
related to proximity to sea ice or the presence of submesoscale activity not well resolved by gridded products.

**Table 2: Comparison of daily satellite-derived velocity to the ITP velocity along the ITP tracks. Correlations (*r*) with *p*<0.05 are in**
**bold. Numbers in the bracket are from the weekly mean timeseries.**

| ITP-Vs | | Mean (ITP) | Mean (Sat.) | STD (ITP) | STD (Sat.) | r | R$^2$ |
|---|---|---|---|---|---|---|---|
| **ITP-70** | Zonal | -0.006 | -0.006 | 0.022 | 0.014 | **0.31 (0.45)** | 0.12 (0.32) |
| | Meridional | -0.005 | -0.003 | 0.021 | 0.015 | **0.42 (0.59)** | 0.20 (0.40) |
| **ITP-77** | Zonal | -0.022 | -0.016 | 0.023 | 0.011 | 0.07 (0.31) | 0.11 (0.20) |
| | Meridional | -0.005 | -0.004 | 0.024 | 0.021 | **0.37 (0.47)** | 0.01 (0.02) |
| **ITP-78** | Zonal | -0.021 | -0.011 | 0.022 | 0.014 | **0.42 (0.78)** | 0.32 (0.77) |
| | Meridional | -0.013 | 0.001 | 0.026 | 0.017 | **0.43 (0.61)** | 0.25 (0.32) |
| **ITP-79** | Zonal | -0.006 | -0.007 | 0.024 | 0.013 | **0.24** (0.38) | 0.02 (0.11) |
| | Meridional | -0.020 | -0.001 | 0.027 | 0.013 | **0.56 (0.76)** | 0.47 (0.60) |
| **ITP-80** | Zonal | -0.004 | -0.001 | 0.020 | 0.012 | **0.42 (0.61)** | 0.30 (0.60) |
| | Meridional | -0.002 | 0.010 | 0.024 | 0.012 | **0.34 (0.52)** | 0.23 (0.16) |
| **Mean** | Zonal | -0.010 | -0.008 | 0.022 | 0.011 | 0.29 (0.50) | 0.13 (0.40) |
| | Meridional | -0.009 | 0.002 | 0.023 | 0.016 | 0.42 (0.59) | 0.22 (0.28) |


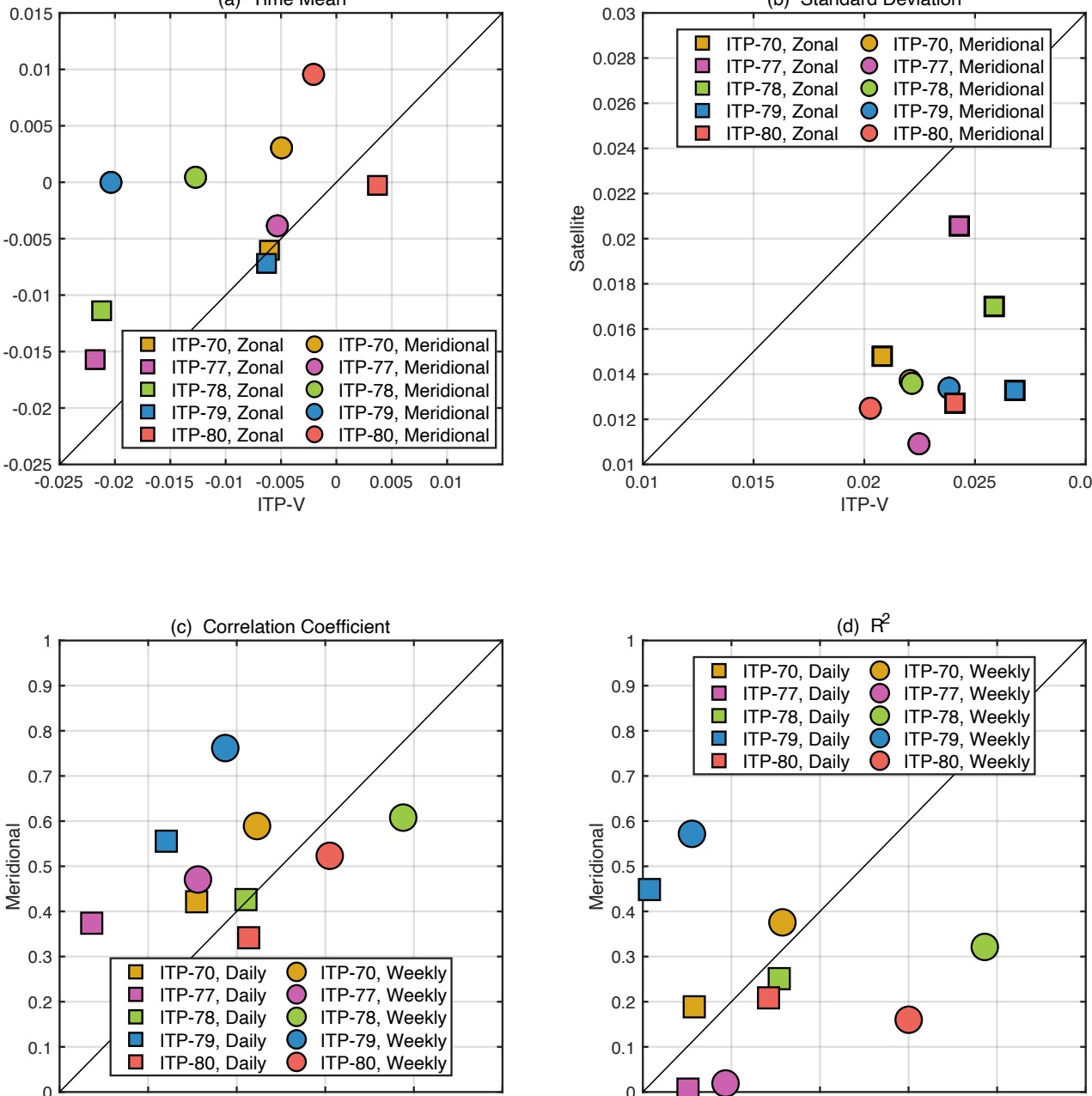


**Figure 14: Scatterplots of statistics of satellite-derived velocity and the ITP velocity. (a) Temporal mean and (b) standard deviation of the paired velocity (unit: m/s). (c) Correlation between satellite-derived velocity and the ITP velocity. (d) Coefficient of determination ($R^2$) of the variation in the ITPs explained by satellite-derived velocity.**


The comparison between satellite-derived ocean surface stress estimates and ITP observations, despite their differences in resolution and sampling, shows encouraging agreement at the first-order level. The satellite stress products successfully capture the broad spatial and temporal variability in surface velocity, supporting the utility of satellite-based estimates in reflecting first-order dynamical signals. This level of agreement supports the overall utility of satellite products in characterizing large-scale stress variability and motivates their continued use in data-sparse polar regions.

However, key limitations remain due to inherent mismatches in spatial and temporal sampling between the datasets. Satellite observations, with their typical resolution of ~25 km and daily sampling frequency, cannot resolve the submesoscale variability and high-frequency processes detectable in the pointwise and often sub-daily ITP profiles (Timmermans et al., 2008). This spatial averaging can smooth out gradients in wind, ice motion, or stress fields that may be sharply defined at smaller scales, particularly in regions such as the marginal ice zone (MIZ) where sea ice concentration and morphology are highly variable (Manucharyan & Thompson, 2017; Alberello et al., 2020).

Temporally, satellite-derived surface stress products may fail to capture transient forcing events such as storm-driven accelerations, inertial oscillations, or short-lived leads in sea ice. In contrast, ITPs can resolve such high-frequency processes (Toole et al., 2011; Timmermans et al., 2012), leading to potential discrepancies when aligning the two datasets. Furthermore, since satellite estimates of stress are often derived from independent wind and ice motion products (e.g., National Centers for Environmental Prediction Reanalysis, NSIDC drift), their accuracy is subject to the limitations of those input fields (Sumata et al., 2014; Lavergne et al., 2010). The lack of fully collocated wind and ice motion fields at the exact time and location of ITP measurements compounds the uncertainty.

These spatial and temporal mismatches further introduce representation errors, as mismatches are not due to sensor or algorithm flaws but rather due to sampling disparities (Janjić et al., 2008). Such errors have been well documented in the context of satellite sea surface salinity (Boutin et al., 2016; Vinogradova et al., 2019). These limitations are evident in the reduced agreement observed for ITP-77, ITP-78, and ITP-79, where several compounding factors likely contributed. First, the ~25 km resolution of the satellite product may be insufficient to resolve submesoscale features and sharp velocity gradients. Second, the timing of deployment in March overlaps with a period of elevated kinetic energy in the Beaufort Gyre (Cassianides et al., 2023), during which intensified eddy activity in the Canada Basin enhances mesoscale variability (Son et al., 2022; Regan et al., 2020). This variability, well captured by the high-resolution ITP profiles, is easily aliased or smoothed out in the satellite-derived daily fields, further amplifying mismatches in direct comparisons.

Moving forward, best practices in validation should account for these differences explicitly. The development of higher-resolution satellite products (Auger et al., 2022; Lucas et al., 2023), along with assimilation into coupled models (Wang et al., 2018), also offers a promising path forward. Increased density of ITP deployments, moored arrays, and coordinated airborne campaigns (Perovich et al., 2023) will be crucial for better spatial coverage in dynamic regions like the Beaufort Gyre and the MIZ.

In conclusion, the comparison reveals that while satellite-derived velocities are subject to systematic biases and reduced variability relative to in situ observations, they nonetheless capture a significant portion of the observed variance, particularly

when considered at weekly timescales. The agreement is stronger for the meridional component and in regions where large-
scale geostrophic flows dominate. These results support the use of satellite-derived velocity products for basin-scale circulation
studies, while also highlighting the need for caution in applications requiring high-frequency or fine-scale flow resolution.
**5 Data Availability**
Daily fields of ocean-surface stress vectors and derived vertical Ekman velocity for the polar oceans are provided for two
periods: 2011–2018 for the Arctic (EPSG number 3408) and 2013–2018 for the Antarctic (EPSG number 3409) and are
available at https://doi.org/10.5281/zenodo.14750492 (Liu & Yu, 2024). The datasets include three auxiliary fields: (i) land
mask, (ii) grid longitudes and latitudes, and (iii) uncertainty estimates for ocean-surface stress.
The input datasets can be found at NSIDC (ice motion: https://nsidc.org/data/nsidc-0116/versions/4; ice extent:
https://nsidc.org/data/nsidc-0051/versions/2) and AVISO website (dynamic topography:
https://www.aviso.altimetry.fr/en/data/products/sea-surface-height-products/regional/arctic-ocean-sea-level-heights.html).
ITP-V data used in this work are retrieved from the WHOI website at https://www2.whoi.edu/site/itp/. CPOM-
DOT/geostrophic currents data are provided by the Centre for Polar Observation and Modelling, University College London
(https://www.cpom.ucl.ac.uk/dynamic_topography). The associated scripts and packages used in this study are openly
available on GitHub at (https://github.com/cydenyliu/Polar_Stress).
**6 Conclusions**
This work presents a daily, 25 km resolution dataset of satellite-derived ocean-surface stress for the Arctic (2011-2018) and
Southern Oceans (2013-2018). The dataset provides detailed daily maps of $\tau_o$ across polar regions north of 60°N and south of
50°S. This dataset achieves finer spatial and temporal resolution, enabling more precise analysis of short-term air-sea
interactions and regional Ekman dynamics. In both the Arctic and Antarctic, it captures short-term and sharp transition between
Ekman upwelling in ice-free regions and downwelling in ice-covered areas.
Uncertainty in the derived ocean-surface stress fields arises primarily from two sources. The first is the spatial filter applied to
the SSH datasets, which reduces small scale variability and enhances consistency between the sea level fields. The second
source of uncertainty is related to the ice-water drag coefficient, which is poorly observed and can vary significantly between
order of $10^{-3}$ and $10^{-2}$. These factors result in a median uncertainty of approximately 20% in the Arctic and about 40% in the
Southern Ocean.
The derived Ekman velocity is used to validate against ITP data from the Arctic's Canada Basin. Satellite-derived surface
velocity, which combine Ekman and geostrophic components, capture over 50% of the observed variation in surface velocity.
Correlation coefficients range from 0.6 to 0.8 on monthly and longer timescales, indicating moderate to strong agreement. It
is important to consider the complex dynamics of the Arctic Ocean when interpreting these statistics. In addition to Ekman

and geostrophic velocity (Regan et al., 2019), processes such as shallow eddy activity (Timmermans et al., 2008; Kenigson et al., 2021; Meneghello et al., 2021), turbulent mixing (Guthrie et al., 2013; Kawaguchi et al., 2014, 2019), and internal waves (Kawaguchi et al., 2016; Zhao et al., 2016) also contribute to the observed variability. Many of these processes remain challenging to observe and parameterize.

Future updates will focus on two primary areas. We plan to extend the dataset's temporal coverage through 2021 by incorporating updated versions of OAFlux and other relevant data products as they become available. This will ensure consistency across components while maintaining the dataset's reliability. Second, the availability of reliable surface height products for the polar region will further enhance data accuracy. While awaiting these advancements, we will assess the potential impacts of transitioning to reanalysis data on our results. Additionally, future research will address key processes that remain underrepresented, such as variable Ekman depth and mesoscale turbulence, to refine the depiction of polar ocean dynamics. Incorporating these factors will improve the ability to capture localized features critical for understanding air-ice-ocean interactions.

## Appendix A: Glossary of Terminology

**Table A1: glossary of terminology and acronyms used in this study.**

| Terminology/Acronyms | Description |
|---|---|
| EASE | Equal-Area Scalable Earth grid |
| SSH | Sea Surface Height |
| OAFlux | Objectively Analyzed Air-Sea Fluxes |
| NSIDC | National Snow and Ice Data Center |
| MEaSUREs | Making Earth System Data Records for Use in Research Environments |
| COARE | Coupled Ocean–Atmosphere Response Experiment |
| SSM/I | Special Sensor Microwave/Imager |
| AMSR-E | Advanced Microwave Scanning Radiometer for EOS |
| AVHRR | Advanced Very High Resolution Radiometer |
| MODIS | Moderate Resolution Imaging Spectroradiometer |
| QuikSCAT | Quick Scatterometer |
| IABP | International Arctic Buoy Programme |
| CLS/PML | Collecte Localisation Satellites / Plymouth Marine Laboratory |
| SMMR | Scanning Multichannel Microwave Radiometer |

| | |
|---|---|
| DMSP | Defense Meteorological Satellite Program |
| SSM | Special Sensor Microwave |
| I-SSMIS | Improved Special Sensor Microwave Imager/Sounder |
| RMSD | Root Mean Square Deviation |
| CPOM | Centre for Polar Observation and Modelling |
| DOT | Dynamic Ocean Topography |
| SIZ | Seasonal Ice Zone |
| PIZ | Perennial Ice Zone |
| MIZ | Marginal Ice Zone |
| STD | Standard Deviation |
| R² | Coefficient of Determination |
| ITP | Ice-Tethered Profiler |

605

## Author contributions

CL: conceptualization, data curation, formal analysis, methodology, software, visualization, writing – original draft preparation, writing – review and editing. LY: conceptualization, project administration, supervision, validation, writing – review and editing.

## Competing interests

The contact author has declared that none of the authors has any competing interests.

## Finance Support

This project is supported by NASA under grant no. 80NSSC23K0981.

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
