# Peer review of "Satellite-based Analysis of Ocean-Surface Stress across the Ice-free and Ice-covered Polar Oceans"

_Earth System Science Data, 2025_

## Author Comment (AC1)

RC1: 'Comment on essd-2025-14', Anonymous Referee #1, 19 Apr 2025
Revised Review of Satellite-based Analysis of Ocean-Surface Stress across the Ice-free and Ice-covered Polar Oceans by Liu and Yu (2025)

The authors present an analysis of satellite-derived ocean-surface stress across the ice-free and ice-covered polar oceans. They utilize a bulk parameterization approach, combining multiple satellite datasets, to estimate ocean-surface stress and Ekman transport. The study provides valuable insights into the spatial and temporal variability of these key drivers of polar climate dynamics. However, I have several concerns regarding the methodology, data processing, and presentation, which must be adequately addressed before the manuscript is suitable for publication.

Thank you for the constructive and thoughtful feedback. In the revised manuscript, we hope that these revisions fully address the reviewer's concerns and strengthen the manuscript and data integrity.

General Comments:

The authors need to update their references to include some relevant papers especially when it relates to relevant dataset (CPOM extended dataset was recently used in Lin et al (2023)), parameterisations (Brenner et al 2021, Lupkes et al 2015), modelling work that can server as motivation (Tsamados et al 2014, Sterlin et al 2023).

Thanks for the suggestion. We have revised the manuscript to include recommended references that align with the datasets, parameterizations, and modeling work discussed in our study (e.g., Line 41-62, 176-185). These additions improve the scientific grounding and contemporary relevance of our study.

The validation with ITP observations is a valuable component, but its limitations need to be discussed in more detail. In particular more discussion is needed and this section needs rewriting to better account for the impact of time and space sampling in the input datasets on the derived ocean surface stress.

Thank you for highlighting the importance of addressing the limitations of the ITP validation. We have revised the relevant section to include a more detailed discussion on the spatial and temporal sampling mismatches between the ITP profiles and the gridded input datasets used to derive ocean surface stress (Line 409-424, 440-562).

Specifically, we now discuss how the relatively coarse resolution of satellite-derived surface stress fields (~25 km, daily) can introduce uncertainties when compared with pointwise and irregularly sampled ITP measurements, particularly in dynamically variable regions. (line 526-562)

These updates clarify the role and interpretation of the ITP validation in our study and better define the bounds within which conclusions can be drawn from the comparison.

The authors perform a sensitivity analysis to value of drag and gaussian spatial filters but do not discuss errors due to model parameterisation (i.e. impact of ue vs ug vs nothing), impact of uncertainty of reanalysis, impact of ocean (and atmospheric) stratification (see papers by Lupkes and Brenner for example), impact of time and length scale over which stresses are calculated.

Thank you for this constructive comment.

We agree that the representation of ocean surface stress is highly sensitive to the velocity assumptions used in parameterizing the drag term. The role of different model parameterization—using surface Ekman velocities ($u_e$), geostrophic velocities ($u_g$), and zero ocean velocity—has been extensively discussed in previous literatures. For example, Zhong et al. (2018) found that mean Beaufort Sea Ekman pumping rates range from -2 to over -10 cm/day depending on whether $u_g$ is included. To strengthen this discussion, we have added references to relevant prior work that explores the impact of velocity choices on Arctic surface stress/Ekman pumping estimation, including Zhong et al., (2018); Ma et al., (2017), and Wu et al. (2021) (Line 101-107).

The discussion on the uncertainty regarding the reanalysis products' uncertainty are extended (line 379-402). We have also included a discussion regarding stratification and how it may alter the coupling between the surface stress and the subsurface response. While our approach assumes a vertically homogeneous Ekman layer, Arctic conditions often exhibit strong vertical stratification—especially in summer—potentially weakening or modifying the expected Ekman velocity profiles.

Finally, we have added text to highlight that our analysis, based on temporally and spatially smoothed fields, may overlook high-frequency variability or sub-gridscale phenomena (Line 526-562). The smoothing scale represents a trade-off between noise suppression and physical fidelity, but we now emphasize the implications of this filtering on stress estimates and Ekman responses, particularly in dynamically active regions or near the ice edge.

We hope these additions clarify the assumptions and limitations of our approach and appropriately contextualize our results within the current literature.

The authors could consider extending the analysis period to include more recent available data (CPOM or CLS) and by removing their current constraint due to the reanalysis they use. Please justify why this reanalysis is really so good to justify loosing 5+ years of analysis.

We chose the CLS dataset over CPOM for sea level primarily due to its higher effective resolution, which is more consistent with the spatial scale (~25 km) of the reanalysis products used, including surface winds from OAFlux2. While CPOM offers a longer record, particularly in the Arctic, its lower effective resolution introduces aliasing and scale mismatches that complicate direct stress estimation (see Auger et al., 2022).

The starting year of 2011 (Arctic) and 2013 (Antarctic) corresponds to the availability of reliable CLS-derived sea level fields. The upper bound of 2018 is constrained by on-going development and refinement of the OAFlux2 wind product. However, we view this analysis not as bound to these

years, but as a scalable framework. All components—wind, ice motion, and sea level—can be updated as newer, equally resolved and validated products become available. We hope this clarifies our choice to prioritize reliability and consistency over extending the time series in this initial effort.

The authors use open access satellite products but fail to share their code! Share code / Github

Thanks for pointing out. All associated scripts and packages have been shared through Github (https://github.com/cydenyliu/Polar_Stress).

References:

Lin, Peigen, Robert S. Pickart, Harry Heorton, Michel Tsamados, Motoyo Itoh, and Takashi Kikuchi. "Recent state transition of the Arctic Ocean's Beaufort Gyre." Nature Geoscience 16, no. 6 (2023): 485-491.

Brenner, Samuel, Luc Rainville, Jim Thomson, Sylvia Cole, and Craig Lee. "Comparing observations and parameterizations of ice-ocean drag through an annual cycle across the Beaufort Sea." Journal of Geophysical Research: Oceans 126, no. 4 (2021): e2020JC016977.

Lüpkes, Christof, and Vladimir M. Gryanik. "A stability-dependent parametrization of transfer coefficients for momentum and heat over polar sea ice to be used in climate models." Journal of Geophysical Research: Atmospheres 120, no. 2 (2015): 552-581.

Tsamados, Michel, Daniel L. Feltham, David Schroeder, Daniela Flocco, Sinead L. Farrell, Nathan Kurtz, Seymour W. Laxon, and Sheldon Bacon. "Impact of variable atmospheric and oceanic form drag on simulations of Arctic sea ice." Journal of Physical Oceanography 44, no. 5 (2014): 1329-1353.

Sterlin, Jean, Michel Tsamados, Thierry Fichefet, François Massonnet, and Gaia Barbic. "Effects of sea ice form drag on the polar oceans in the NEMO-LIM3 global ocean–sea ice model." Ocean Modelling 184 (2023): 102227.

Thank you for the suggestions. All references are incorporated in the revised manuscript.

Specific Comments:

L6 EASE2? clarify if so

EASE2 stands for Equal-Area Scalable Earth (EASE) Grid 2.0. It is introduced in the abstract (line 7) and introduction (line 73). In the revised manuscript it is explained explicitly (line 139).

L15 clarify why you need both CPOM and CLS?

Both the CLS/PML multi-altimeter Arctic sea level dataset and the CPOM Dynamic Ocean

Topography (DOT) dataset are utilized in this study to leverage their complementary strengths in capturing Arctic Ocean dynamics.

The CLS dataset offers high-resolution (25 km) sea level anomaly maps every 3 days, derived from a combination of SARAL/AltiKa, CryoSat-2, and Sentinel-3A altimetry missions. This multi-mission approach enhances spatial and temporal coverage, providing detailed insights into mesoscale features and reducing spurious signals often associated with single-mission datasets (line 139-143).

Conversely, the CPOM DOT dataset provides a longer temporal record (2003–2021) with monthly resolution, offering valuable context for interannual and decadal variability studies. Its extended coverage is particularly useful for assessing long-term trends in Arctic sea level and circulation patterns (176-185).

By combining these datasets, the study benefits from the high-resolution, short-term variability captured by CLS and the long-term trends provided by CPOM. This integrated approach allows for a more comprehensive analysis of Arctic Ocean surface stress and circulation dynamics.

To reconcile the differing resolutions and temporal scales, a 2D Gaussian filter with a standard deviation of 75 km is applied, ensuring consistency and interpretability between the datasets. Sensitivity tests confirm that this filtering approach effectively balances the preservation of mesoscale features with the reduction of noise, facilitating robust comparisons and analyses. (line 165-185)

L19 clarify / correlation of what

It is revised as "Validation efforts using Ice-Tethered Profiler velocity records revealed weak to moderate correlations with satellite-derived stress (r = 0.4–0.8) between observed surface velocities and satellite-derived estimates (Ekman + geostrophic) at daily resolution, with significantly improved agreement when averaged to weekly means." (Line 18-20).

L21 not showed in this paper?

Thanks for pointing out. It has been removed from the abstract.

L40 ice-ocean governor is Menenghello citation?

The correct reference (Meneghello et al., 2017) is added (Line 39).

L47 add references here i.e Lin et al 2022 but also more references in introduction for motivation / modelling work

The motivation section is revised at line 41-62. Lin et al., 2022 is also referenced at line 181.

L51 Seems a waste to limit yourselves to 2018 due to reanalysis

Please see our response to the earlier comment for further clarification.

Here it is explicitly revised as "The analysis period ends in 2018 to maintain consistency with the most reliable iteration of the ongoing refinement of OAFlux wind product. While this choice limits the temporal extent, the framework itself remains flexible and can be readily extended as newer, better-resolved datasets become available." (line 150-152).

L77 cite recent papers (including modelling)

We believe it is more appropriate to address this point elsewhere for better alignment with the overall structure of the manuscript. Accordingly, we have incorporated the relevant references as motivation at Line 50-62.

L92 discuss impact on ocean surface stress of time sampling i.e. daily vs 3d vs monthly

Thanks for the suggestion. A discussion is added at line 165-175.

L92 discuss why this reanalysis so important

We now clarify that the CPOM DOT dataset is incorporated as a reference due to its wider usage and demonstrated utility in capturing Arctic-scale variability, despite its lower temporal and spatial resolution. In contrast, the CLS product offers finer-scale structure but can be harder to validate in data-sparse polar regions. To balance these strengths, we apply spatial filtering to the CLS fields, allowing consistent comparison and improved interpretability.

A discussion is added at line 176-185.

L117 add references to paper Lin that used more extended CPOM extension reference

It is added line 181.

L126 is it worth limiting to that reanalisis then?

Please see our response to the earlier comment for L51.

Figure 2 why gaussian filtering to reconcile with CPOM?

Please see our response to the earlier comment for L92.

L144 why not just use CLS if better? You criticise CPOM product but you fail to discuss all issues in these CLS and CPOM product when it comes to continuity of SSH at the ice-ocean boundary.

Please see our response to the earlier comment for L92.

L186 clarify

It is revised as "The coefficient of determination (R2, here is calculated as the proportion of variance explained by the seasonal cycle, i.e., $R^2 = 1 - \frac{\sum_i(y_i - y_{seasonal})^2}{\sum_i(y_i - \bar{y})^2})$ " (line 229-230).

L190 not so clear on the map. Shouldn't the maps be monthly?

We appreciate this comment. The maps in Figure 4 are indeed time-mean values of the Ekman pumping rate, calculated over the entire study period. It was explicitly clarified in the figure caption that these maps represent time-averaged data, not monthly maps. The decision to present time-mean values was made to highlight broad spatial patterns in Ekman pumping across the study region.

While monthly maps are possible, this approach aligns with the goal of presenting a more general overview, as this paper serves as an introduction to the dataset rather than focusing on seasonality or detailed temporal changes. This also ensures the reader understands the temporal scope of the datasets.

Figure 4 mean is noisy so what about monthly / daily products. Is that mean over all winter and all months?

We appreciate the suggestion regarding the apparent noise in the time-mean pattern shown in Figure 4. We would like to clarify that the "noise" observed is actually fine-scale structure originating from the CLS sea surface height (SSH) product, which is inherently resolved at higher spatial scales. Furthermore, we have applied a spatial filter to mitigate small-scale variations and smooth the pattern to reconcile with the large scale variability captured by CPOM, though the finer features may still be visible in the time-mean pattern.

The mean is calculated across all months, providing a broader view of the Ekman pumping rate's spatial structure. We believe that this method captures important spatial features of the variability, which would be less discernible in individual monthly maps.

Figure 5 what is the total region used? Not clear what is aaw

As stated in the caption of Figure 5, variables are subscripted *aw* when averaged or summed over ice-free open ocean, and *iw* when averaged or summed over ice-covered open ocean. These regions are defined dynamically based on the daily sea ice coverage.

L277 cite recent papers and modelling ones

We believe it is more appropriate to add the discussion at Line 50-62.

L312 what about error from time sampling, model physics parameterization

Thanks for the suggestion. Discussion on errors from other sources are discussed at line 379-402.

Figure 9 is based on two errors only not sure I buy this. Also not clear how the range of tau_w chose 10^-3 to 10^-2.

We appreciate the reviewer's concern and clarify that the two error sources in Figure 9 represent only calculative uncertainty arising from the derivation of surface stress using wind and ice motion inputs. As noted in Line 379-402, additional sources of uncertainty—such as reanalysis forcing variability, stratification effects, and scale-dependent representation errors—are discussed in detail in the manuscript and will be further addressed in future work using ensemble methods and improved parameterizations.

Regarding the range of $\tau\_w$, we emphasize that this range was not selected arbitrarily, but rather calculated directly from the input data over the study period. It reflects the physically realistic surface stress values derived from the observed wind and ice motion fields, consistent with reported stress magnitudes in Arctic conditions (e.g., Ma et al., 2017). These values span typical ice-covered and open-ocean conditions and form the basis for assessing vertical stress-driven processes like Ekman pumping.

L340 test impact on derived stress

We acknowledge the challenge in evaluating derived surface stress due to the lack of direct observations. To assess its robustness, we conducted sensitivity tests exploring the impact of drag parameterizations, spatial filtering. Additionally, we validated the resulting stress indirectly by comparing the derived velocity fields against Ice-Tethered Profiler observations. While these comparisons provide a first-order assessment, further improvements are expected with enhanced understanding of ice–ocean coupling and boundary layer processes in polar conditions.

L366 shouldn't you compare at the time averageds used in stress calculations? i.e. daily

We appreciate the comment. In the revised manuscript, all comparisons have been updated: rather than interpolating the satellite product to match ITP times, we now compute time-averaged ITP velocities (daily and weekly) and compare them directly with the daily satellite-derived velocity fields. This approach improves the temporal consistency and better reflects the resolution limits of the satellite dataset.

We find that the key conclusions remain consistent, though the number of independent points is reduced.

Figure 11 is u really daily?

In the revised version of Figure 11, we compare the daily means of zonal and meridional velocities.

Figure 12 representation issue here clearly seen (space or time or both?)

Thanks for the observation. In the revised manuscript, Figure 12 now uses daily mean fields, which improves consistency between the in situ and satellite-derived datasets. While some spatial and temporal mismatch remains—due to the intrinsic differences between gridded satellite fields and point-based ITP data—we believe the daily averaging better represents the dynamics of interest without introducing excessive smoothing. The representativity limitations related to spatial resolution are discussed further in Lines 526–562.

Figure 14 obvious representativity issue (space, time or both?). I would redo all this section on monthly products and discuss better

We appreciate this suggestion. In the revision, we retain the daily mean products to preserve temporal fidelity and allow clearer interpretation of short-timescale variability. Monthly means were tested but tended to obscure key features relevant to the comparison.

Figure 15 what about plot for daily timescales

In the revised version, the original Figure 15 has been removed, as it did not effectively convey the intended message or align well with the main discussion. Instead, the relevant information has been integrated into the revised Table 3, which now more clearly presents the key statistics and better supports the interpretation of daily variability across the datasets.

L433 share code!

At line 573-576 it is added as "The associated scripts and packages used in this study are openly available on GitHub at (https://github.com/cydenyliu/Polar_Stress)."

References:
Auger M, Prandi P, Sallée J B. Southern ocean sea level anomaly in the sea ice-covered sector from multimission satellite observations. Scientific Data, 9(1): 70, 2022.
Wu, Y., Wang, Z. and Liu, C.. Impacts of changed ice-ocean stress on the North Atlantic Ocean: Role of ocean surface currents. Frontiers in Marine Science, 8, p.628892, 2021.
Ma, B., M. Steele, and C. M. Lee: Ekman circulation in the Arctic Ocean: Beyond the Beaufort Gyre. J. Geophys. Res. Oceans, 122, 3358–3374, https://doi.org/10.1002/2016JC012624, 2017.
Meneghello, G., J. Marshall, S. T. Cole, and M.-L. Timmermans: Observational inferences of lateral eddy diffusivity in the halocline of the Beaufort Gyre. Geophys. Res. Lett., 44, 12 331–12 338, 2017.
Zhong, W., Steele, M., Zhang, J. and Zhao, J.. Greater role of geostrophic currents in Ekman dynamics in the western Arctic Ocean as a mechanism for Beaufort Gyre stabilization. Journal of Geophysical Research: Oceans, 123(1), pp.149-165, 2018.

---

## Author Comment (AC2)

**RC2: 'Comment on essd-2025-14', Anonymous Referee #2, 21 Apr 2025**

This is a well written manuscript and the analysis seems legitimate and thoughtful. The first thing that came to me after reading the manuscript is why the new dataset of ocean surface stress is needed? Since there are many assumptions have been made in order to calculate the ocean-surface stress. It will be great if the authors can explain the motivation and reasoning that this data is needed for the community.

Thank you for the thoughtful feedback and for recognizing the quality of the analysis. We agree that a clearer articulation of the motivation behind generating this dataset is important, and we have now made this more explicit in the revised manuscript.

Surface stress is a fundamental driver of polar ocean circulation, as it governs the transfer of momentum from the atmosphere to the ocean—either directly or through the mediating influence of sea ice. In the Arctic, this stress regulates key processes including ice drift, Ekman transport, and the spin-up of gyre systems. Recent studies have shown that thinning and retreating sea ice can lead to enhanced wind–ocean coupling, with significant implications for ocean kinetic energy, vertical mixing, and climate feedbacks (e.g., Martin et al., 2013; Martin et al., 2016; Muilwijk et al., 2024). Despite its importance, surface stress remains poorly constrained in both observations and models, especially in the ice-covered ocean where coupled feedbacks are complex and highly variable.

To address this gap, our dataset offers an observationally anchored, spatially continuous estimate of surface stress across the Arctic and Antarctic, separating ice-covered and open-ocean contributions using consistent and transparent assumptions. While simplifications (e.g., fixed drag coefficients, Gaussian filters) are necessary, they are based on recent literature and can be adapted as new data or parameterizations become available.

This product is intended not as a replacement for high-resolution coupled models, but as a complementary tool for:

Forcing ocean models where stress observations are sparse or missing;

Benchmarking atmosphere-ice-ocean coupling in simulations;

Quantifying trends in stress-driven processes such as gyre intensification or ice divergence.

We have added dedicated paragraphs in the Introduction to better articulate these motivations (line 41-62).

Second, the value of Ekman depth used in the calculation is not explained, eqn. 4 & 5. The Ekman depth is needed in order to calculate the vertical Ekman velocity. What are the values of Ekman depth that you used?

Thanks for this observation. The Ekman depth used in our calculations is set to 20 m following Meneghello et al. (2018), and this is stated at line 112. We chose this fixed value based on typical

estimates from prior observational and modeling studies in polar regions (e.g., Zhong et al., 2018; Ramadhan et al., 2022).

We fully acknowledge that the Ekman depth can vary significantly depending on stratification, ice cover, and wind forcing. This introduces an important source of uncertainty in the derived vertical velocity, which we plan to explore in future work.

Third, for the scatterplots of ITP vs satellite (Fig. 12 & 13), how the ITP data is processed to be comparable to the daily satellite product? Did you do time average on ITP data to daily data or interpolate the satellite daily product to the ITP profile time? From the number of samples, it seems the author interpolated the daily satellite data to the ITP data (profile) time. If so, the scatter plots are meaningless. The comparison is wrong, since the satellite temporal resolution is under sampled. You can not draw conclusions from these scatterplots. Also, the 30-day low pass resulted in too few data points (Fig. 14, 15 & Table 3). I suggest to do a time-average ITP data to daily or weekly and , and redo the analysis.

Thank you for highlighting this concern. We agree that differences in temporal sampling between satellite-derived and in situ data must be carefully considered. In our analysis, we interpolated the daily satellite product to the timestamp of the ITP profiles to maximize the number of data pairs, which is a common practice when comparing gridded satellite fields with in situ observations (e.g., Guinehut et al., 2012).

However, we acknowledge that this approach may introduce representativeness errors if the satellite data does not resolve variability at the timescale of the in situ profiles. To address this concern, we have repeated the analysis by averaging the ITP profiles to daily and weekly means, and then comparing these averages to the corresponding satellite product without interpolation (see revised section 4.2, line 403-562). The revised scatterplots and correlation statistics are now shown in updated Figs. 12–15 and Table 3. We find that the key conclusions remain consistent, though the number of independent points is reduced.

**Other small comments**

**Line 40: "ice-ocean governor," $\rightarrow$ "ice-ocean governor",**

**It is revised accordingly (line 39).**

Line 60: Figure 1(a) caption: there is no dashed magenta has shown. Is the September ice extent boundaries out of the scope of the plot? The Solid line near the Novaya Zemlya has a portion of dashed line. Is this supposed to be solid?

Thanks for pointing out. In the revised Figure 1 we added the missing contours and noted the marginal seas.

Line 84 It is unclear to me how the De: Ekman layer depth is determined in the calculation. Can you

**explain?**

In the revision, we now explicitly define the Ekman layer depth as 20 m at line 112, following previous studies such as Meneghello et al. (2018).

Acronym stands for? Line 98 acronym OAFLux2? Line 106 acronym SSM/I, AMSR-E, AVHRR, MODIS, QuikSCAT? Line 107 acronym IABP? Line 111 acronym CLS/PML? Line 120 acronym SMMR DMSP, NSIDC?

In response to these comments, a table of acronyms has been added as an appendix to the manuscript to address above comments (see line 603).

Line 139 "Noting the 25 km resolution could introduce uncertainties near the 15% sea ice concentration boundary." --> How did you arrive this conclusion?

We thank the reviewer for this question. This statement is based on the fact that a 25 km resolution grid cell can encompass both ice-covered and open water areas near the 15% sea ice concentration threshold, which is commonly used in satellite products. At this resolution, subgrid-scale variability becomes significant, and the true ice edge may not be well resolved—potentially affecting ice classification and derived variables such as surface stress. This issue has been previously noted in the literature (e.g., Meier, 2005; Ivanova et al., 2015) and is particularly relevant in the marginal ice zone where gradients in sea ice concentration are sharp.

In the revised text "Noting that the 25 km resolution may introduce uncertainties near the 15% sea ice concentration boundary, as such coarse resolution can obscure sharp gradients in the marginal ice zone and misclassify mixed ice–water grid cells (e.g., Meier, 2005; Ivanova et al., 2015)" (line 162-164).

Line 152 "where W represents the local vertical Ekman velocity  $W_e" \rightarrow W$  and  $W_e$  were used interchangeably. Can you decide which one to use and keep it consistently?

Thank you for pointing this out. We have revised the manuscript to use W\_e at line 194-195, to avoid any confusion.

Line 227 "in the Indian Ocean sector and the southeast Pacific"  $\rightarrow$  Can you label it on Fig 6b?

Thank you for the suggestion. We have revised the text at Line 270 to specify the approximate longitude ranges instead of using regional names. This improves clarity and avoids ambiguity.

Line 232 of 50°S (Figure 6d),  $\rightarrow$  Can you label the latitude and longitude on Fig 6? Line 293 Baffin Bay, the Chukchi Sea, and north of Fram Strait  $\rightarrow$  Can you label these places in Figure 8?.

Lien 324: the Mendeleev Ridge.  $\rightarrow$  please label it on the Figures Line 327-328 Weddell and Ross Seas, Antarctic Peninsula and western Ross Sea, Enderby Land and the Amundsen Sea  $\rightarrow$  please label these places on the Figures.

In response to your above comments, the geographic maps in Figure 1 have been revised to include the names of key geographic locations and marginal seas to improve clarity and orientation (line 80). We note that adding extensive labels or coordinate grids to the main result figures may hinder readability, so detailed labeling is limited to overview maps for reference.

**Line 309 What do you mean "full eight-year/six-year period" period?**

The phrase "full eight-year/six-year period" refers to the different temporal coverage used for each region, and has been revised as "an eight-year period (2011–2018) for the Arctic, and a six-year period (2013–2018) for the Antarctic" (Line 351-352).

**Line 381 What do you mean "paired" observations?**

By 'paired' observations, we refer to data points that are co-located in both space and time. This has been clarified in the revised text as: 'comparison of satellite-derived surface velocity components against collocated ITP-V observations' (line 455).

Line 385 "ITP-80 pair shows better agreement in the zonal component after 200 days."  $\rightarrow$  I can't tell if this is true from the figure 12. How did you arrive this conclusion?

Thank you for pointing this out. We agree that the statement was ambiguous and not clearly supported by Figure 12. To avoid potential confusion, this sentence has been removed in the revised version.

References:

Ramadhan A, Marshall J, Meneghello G, et al. Observations of upwelling and downwelling around Antarctica mediated by sea ice. Frontiers in Marine Science, 9: 864808, 2022.

Guinehut, S., Dhomps, A.L., Larnicol, G. and Le Traon, P.Y.. High resolution 3-D temperature and salinity fields derived from in situ and satellite observations. Ocean Science, 8(5), pp.845-857, 2012. Meneghello, G., J. Marshall, S. T. Cole, and M.-L. Timmermans: Observational inferences of lateral eddy diffusivity in the halocline of the Beaufort Gyre. Geophys. Res. Lett., 44, 12 331–12 338, 2017. Meneghello, G., Marshall, J., Timmermans, M. L., & Scott, J.: Observations of seasonal upwelling and downwelling in the Beaufort Sea mediated by sea ice. Journal of Physical Oceanography, 48(4), 795-805, 2018.

Zhong, W., Steele, M., Zhang, J. and Zhao, J.. Greater role of geostrophic currents in Ekman dynamics in the western Arctic Ocean as a mechanism for Beaufort Gyre stabilization. Journal of Geophysical Research: Oceans, 123(1), pp.149-165, 2018.

Martin, T., Steele, M. and Zhang, J., Seasonality and long-term trend of Arctic Ocean surface stress in a model. Journal of Geophysical Research: Oceans, 119(3), pp.1723-1738. 2014.

Martin, T., Tsamados, M., Schroeder, D. & Feltham, D. L. The impact of variable sea ice roughness on changes in arctic ocean surface stress: A model study. J. Geophys. Res.: Oceans 121, 1931–1952, 2016.

Muilwijk, M., Hattermann, T., Martin, T. and Granskog, M.A.. Future sea ice weakening amplifies wind-driven trends in surface stress and Arctic Ocean spin-up. Nature Communications, 15(1), p.6889, 2024.

Ivanova, N., Pedersen, L.T., Tonboe, R.T., Kern, S., Heygster, G., Lavergne, T., Sørensen, A., Saldo, R., Dybkjær, G., Brucker, L.J.T.C. and Shokr, M.. Inter-comparison and evaluation of sea ice algorithms: towards further identification of challenges and optimal approach using passive microwave observations. The Cryosphere, 9(5), pp.1797-1817, 2015.

Meier, W.N.. Comparison of passive microwave ice concentration algorithm retrievals with AVHRR imagery in Arctic peripheral seas. IEEE Transactions on geoscience and remote sensing, 43(6), pp.1324-1337, 2005.